# Hierarchical Koopman Diffusion: Fast Generation with Interpretable Diffusion Trajectory

**Hanru Bai**
Fudan University
hrbai23@m.fudan.edu.cn

**Weiyang Ding***
Fudan University
dingwy@fudan.edu.cn

**Difan Zou**
The University of Hong Kong
dzou@cs.hku.hk

## Abstract

Diffusion models have achieved impressive success in high-fidelity image generation but suffer from slow sampling due to their inherently iterative denoising process. While recent one-step methods accelerate inference by learning direct noise-to-image mappings, they sacrifice the interpretability and fine-grained control intrinsic to diffusion dynamics, key advantages that enable applications like editable generation. To resolve this dichotomy, we introduce **Hierarchical Koopman Diffusion**, a novel framework that achieves both one-step sampling and interpretable generative trajectories. Grounded in Koopman operator theory, our method lifts the nonlinear diffusion dynamics into a latent space where evolution is governed by globally linear operators, enabling closed-form trajectory solutions. This formulation not only eliminates iterative sampling but also provides full access to intermediate states, allowing manual intervention during generation. To model the multi-scale nature of images, we design a hierarchical architecture that disentangles generative dynamics across spatial resolutions via scale-specific Koopman subspaces, capturing coarse-to-fine details systematically. We empirically show that the Hierarchical Koopman Diffusion not only achieves competitive one-step generation performance but also provides a principled mechanism for interpreting and manipulating the generative process through spectral analysis. Our framework bridges the gap between fast sampling and interpretability in diffusion models, paving the way for explainable image synthesis in generative modeling.

## 1 Introduction

Diffusion models [29, 7, 30] have achieved remarkable success in image generative tasks [3]. However, despite producing high-fidelity samples, the sampling process of diffusion models typically requires an expensive iterative procedure, which limits their applicability in real-world scenarios [18]. Accelerating sampling in diffusion models thus remains a critical challenge. To overcome this limitation, recent work has focused on efficient one-step inference for diffusion models to replace the costly iterative sampling process.

Existing approaches toward this goal fall into several distinct paradigms. A widely adopted paradigm is distillation-based methods [39, 42], which aim to distill pre-trained diffusion models into efficient one-step generators. Among these, Knowledge Distillation (KD) [17], Progressive Distillation (PD) [27], and (distilled) Rectified Flow [14, 13] are considered classical approaches, laying the foundation

---

*Corresponding author

39th Conference on Neural Information Processing Systems (NeurIPS 2025).

for recent advances in one-step generation. Another influential direction involves consistency models [32, 31], which learn time-consistent mappings from noisy inputs to clean data.

Although these methods achieve strong performance in one-step image generation, they fundamentally rely on learning direct noise-to-image mappings, bypassing the temporally coherent denoising trajectory upon which diffusion models are built. As a result, they lack access to intermediate generative states of the diffusion trajectory that are crucial for interpreting and controlling sample evolution from noise to image, thus leading to limited interpretability of the generation process. This lack of interpretability not only obscures how semantic and structural information gradually emerges during sampling but also limits the ability to intervene at specific stages along the generative trajectory—an ability that enables controllable image synthesis at inference time. [2]

Our goal is to develop an explicitly interpretable one-step generation framework, which is not merely to achieve high generation quality in a single step, but to provide a principled understanding of the generative process and thus enable fine-grained control along the diffusion trajectory. Motivated by Koopman operator theory [20, 24], which projects the nonlinear dynamics into an observable space where evolution is linear via function space lifting, we propose an *explicit* one-step generation paradigm that mapping the entire deterministic sampling trajectory of diffusion models into a latent space where closed-form ODE solutions exist. This latent space, theoretically grounded in Koopman theory, is referred to as the Koopman space. In this space, the dynamics of the diffusion process are governed by a linear operator, thus revealing an explicit, analytically tractable form for the evolution to enable an interpretable mapping from noise to data. We realize this framework by jointly learning the mapping and its associated linear operator. In our explicit framework, all intermediate states along trajectories are analytically accessible, thus naturally allowing for introducing additional supervision on these states to guide the noise-to-image mapping more precisely and enabling fine-grained control of the generation dynamics, a capability not afforded by implicit methods.

While the Koopman-based modeling provides a principled way to enable explicit one-step sampling, its standard formulations operate in a single latent space, implicitly assuming uniform dynamics across all spatial and semantic scales. However, generative processes, particularly in visual domains, exhibit inherently multi-scale behaviors: global structures emerge early and evolve slowly, while fine-grained textures form later with more rapid variations [34]. Ignoring this scale-specific nature limits the model's capacity to represent the full complexity of image generation. To address this limitation, we reformulate a *hierarchical* Koopman modeling framework that explicitly decomposes the generative dynamics across multiple spatial resolutions. Features at different scales are extracted via a U-Net encoder and projected into separate Koopman subspaces, each governed by its own linear operator. This design allows the model to track the evolution of visual content from coarse layout to fine detail, aligning with the hierarchical nature of visual perception and offering improved generation fidelity. Our key contributions are summarized as follows:

- We propose a novel interpretable one-step generation paradigm, referred to as Hierarchical Koopman Diffusion (HKD), via hierarchical Koopman dynamics. This paradigm uniquely integrates explicit intermediate generative states into the process, enabling control along the diffusion trajectory. Moreover, our framework provides a novel analytical aspect for diffusion models using spectral tools in dynamical systems theory to analyze the underlying generative mechanisms.

- We provide a theoretical justification that our Koopman explicit formulation is provably more expressive than directly learning a black-box mapping from noise to image using standard neural networks for learning the diffusion process.

- We conduct experiments on the CIFAR-10 and FFHQ datasets to demonstrate the competitive one-step generation performance of our proposed framework. Beyond generation quality, we interpret the underlying generative dynamics via principled spectral analysis, revealing a quantitative correspondence between spectral components and semantic image attributes. Moreover, we further justify the interpretability of our framework through an image editing experiment that targets frequency-specific intervention at the intermediate stage of the diffusion trajectory.

---

[2]Some distillation-based fast generation methods have explored multi-step editing for text-to-image, which is beyond the scope of this work. We focus on trajectory-level intervention during one-step generation.

## 2 Background

**Diffusion Models.** We consider the continuous-time diffusion models [33], where data is generated by reversing a stochastic process that gradually adds Gaussian noise. Let $p_{\text{data}}(\boldsymbol{x})$ denote the data distribution. The diffusion process is described by the stochastic differential equation (SDE): $\mathrm{d}\boldsymbol{x}_t = \boldsymbol{\mu}(\boldsymbol{x}_t, t)\mathrm{d}t + \sigma(t)\mathrm{d}\boldsymbol{w}_t$, where $t \in [0, T]$, $\boldsymbol{\mu}(\cdot, \cdot)$ and $\sigma(\cdot)$ are the drift and diffusion coefficients, respectively. $\{\boldsymbol{w}_t\}_{t \in [0,T]}$ represents standard Brownian motion. The distribution of $\boldsymbol{x}_t$ is $p_t(\boldsymbol{x})$, with $p_0(\boldsymbol{x}) \equiv p_{\text{data}}(\boldsymbol{x})$. Notably, there exists an ordinary differential equation (ODE), called the Probability Flow ODE, whose solution trajectories sampled at $t$ follow $p_t(\boldsymbol{x})$:

$$\mathrm{d}\boldsymbol{x}_t = \left[\boldsymbol{\mu}(\boldsymbol{x}_t, t) - \tfrac{1}{2}\sigma(t)^2 \nabla \log p_t(\boldsymbol{x}_t)\right] \mathrm{d}t, \tag{1}$$

where $\nabla \log p_t(\boldsymbol{x}_t)$ is the score function of $p_t(\boldsymbol{x}_t)$.

**Koopman Operators.** Koopman theory provides a framework for analyzing nonlinear dynamical systems by transforming them into a linear form. The formal definition of the Koopman operator is given below.

**Definition 2.1** (Koopman Operator [21]). *Consider a finite-dimensional state space $\mathcal{X} \subseteq \mathbb{R}^n$ with state evolution described by $\boldsymbol{x}_{t+\Delta t} = \Phi(\boldsymbol{x}_t), t \in [t_0, t_1]$, where $\Phi : \mathcal{X} \to \mathcal{X}$ is the state transition operator. The Koopman operator $\mathcal{K}$ is a linear operator that acts on an infinite-dimensional space of observable function $g : \mathbb{R}^n \to \mathbb{R}$, such that*

$$(\mathcal{K} \circ g)(\boldsymbol{x}) = g(\Phi(\boldsymbol{x})). \tag{2}$$

Instead of modeling the state $\boldsymbol{x}_t$ directly, the Koopman operator captures the evolution of observables $\boldsymbol{z}_t = \boldsymbol{g}(\boldsymbol{x}) \triangleq [g_1(\boldsymbol{x}), \cdots, g_m(\boldsymbol{x})]^\top$ in a lifted space, where each $g_i$ is a component of the observable function **g**. In continuous time, this yields a linear system $\mathrm{d}\boldsymbol{z}_t/\mathrm{d}t = \boldsymbol{A}\boldsymbol{z}_t$, enabling spectral analysis of nonlinear dynamics. Formal details on approximating with finite observables are in App. A.2.

## 3 Hierarchical Koopman Diffusion for Fast Generation

We propose a novel denoising generative model called Hierarchical Koopman Diffusion (HKD) that enables one-step deterministic sampling for diffusion models via Koopman operator theory. By leveraging the Koopman operator's ability to use tools of linear dynamics such as spectral analysis, our formulation not only enables efficient one-step ODE sampling but also offers a principled dynamical interpretation of generative modeling. This section is organized as follows: we first introduce the theoretical formulation of our framework (Sec. 3.1), then describe the corresponding supervision strategy (Sec. 3.2), and conclude with implementation details (Sec. 3.3). We subsequently present theoretical results that justify the superiority of the proposed method under ideal conditions (Sec. 3.4).

### 3.1 Framework Formulation

**Encoder.** Given the trajectory $\{\boldsymbol{x}_t\}_{t \in [\epsilon, T]}$ satisfing ODE in Eq. (1), we construct a hierarchical Koopman representation by applying an encoder

$$\mathcal{E}_{\boldsymbol{\theta}} : \boldsymbol{x}_t \in \mathbb{R}^{C \times H \times W} \mapsto \big\{\boldsymbol{z}_t^{(l)} \in \mathbb{R}^{d_l \times h_l \times w_l}\big\}_{l=1}^{L}.$$

$\mathcal{E}_{\boldsymbol{\theta}}$ adopts a U-Net style downsampling architecture, just as the right part of Fig. 1. Here, $d_l$ is the number of latent channels and $(h_l, w_l)$ is the image resolution at level $l$. The encoder approximates independent Koopman observable functions $\boldsymbol{g}^{(l)} : \mathbb{R}^{C \times H \times W} \to \mathbb{R}^{d_l \times h_l \times w_l}$ at each level by obtaining $\boldsymbol{z}_t^{(l)} \approx \boldsymbol{g}^{(l)}(\boldsymbol{x}_t)$, where $\boldsymbol{g}^{(l)}$ captures the dynamics corresponding to its spatial scale.

**Hierarchical Koopman Subspace.** Each latent feature $\boldsymbol{z}_t^{(l)} \in \mathbb{R}^{d_l \times h_l \times w_l}$ is extracted at level $l$, with $\boldsymbol{z}_t^{(l)}(i, j) \in \mathbb{R}^{d_l}$ denoting the feature vector at spatial position $(i, j)$. We model $\boldsymbol{z}_t^{(l)}(i, j)$ evolves linearly under a spatially-varying linear operator $\boldsymbol{A}^{(l)}(i, j) \in \mathbb{R}^{d_l \times d_l}$, which is specific to each spatial location $(i, j)$, just as the middle part of Fig. 1:

$$\tfrac{\mathrm{d}\boldsymbol{z}_t^{(l)}(i,j)}{\mathrm{d}t} = \boldsymbol{A}^{(l)}(i, j)\, \boldsymbol{z}_t^{(l)}(i, j), \ \forall(i, j), \ t \in [\epsilon, T]. \tag{3}$$

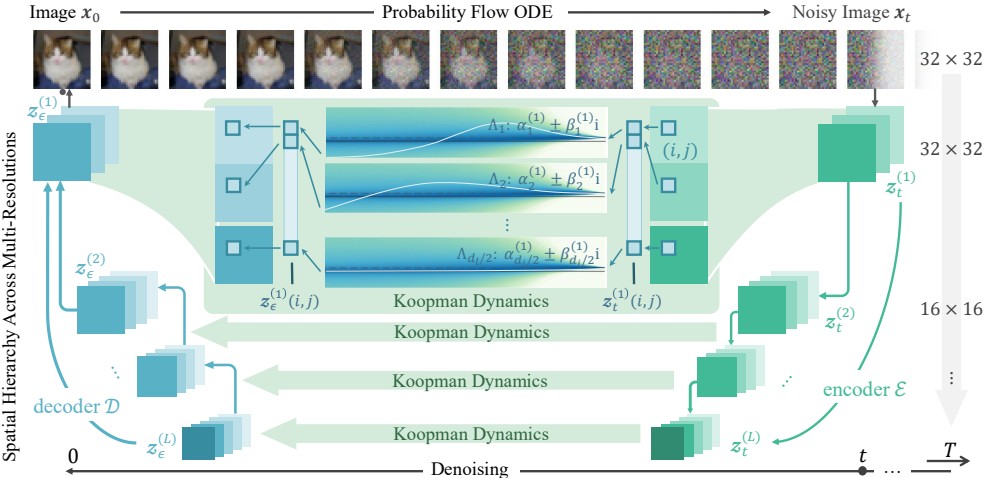

Figure 1: The framework of the proposed method. The HKD model first hierarchically extracts different-level features from the given noisy image at any time $t$ by encoder $\mathcal{E}$. Secondly, the Koopman dynamics model is applied for each level to the skips and the bottleneck. Last, a uniform decoder $\mathcal{D}$ performs the mapping from the Koopman spaces back to the image space.

The latent evolution at each position thus follows an independent linear dynamical system governed by its local Koopman operator. Allowing $\boldsymbol{A}^{(l)}(i,j)$ to vary across spatial locations enables the model to capture heterogeneous temporal-frequency behaviors at different regions, thus leading to finer-grained and spatially adaptive dynamics modeling.

Without loss of generality, $\boldsymbol{A}^{(l)}(i,j)$ could be modeled in the form of block-diagonalizable, i.e.,

$$\boldsymbol{A}^{(l)}(i,j) = \text{diag}\{\boldsymbol{\Lambda}_1^{(l)}(i,j), \boldsymbol{\Lambda}_2^{(l)}(i,j), \cdots, \boldsymbol{\Lambda}_k^{(l)}(i,j), \cdots, \boldsymbol{\Lambda}_{d_l/2}^{(l)}(i,j)\} \tag{4}$$

with $d_l$ even and each block $\boldsymbol{\Lambda}_k^{(l)}(i,j) \in \mathbb{R}^{2\times2}$ corresponding to a pair of complex conjugate eigenvalues $\alpha_k^{(l)} \pm i\beta_k^{(l)}$ of $\boldsymbol{A}^{(l)}(i,j)$, given explicitly by $\boldsymbol{\Lambda}_k^{(l)}(i,j) = \begin{bmatrix} \alpha_k^{(l)}(i,j) & \beta_k^{(l)}(i,j) \\ -\beta_k^{(l)}(i,j) & \alpha_k^{(l)}(i,j) \end{bmatrix}$. This reduces the number of parameters compared to modeling a full matrix. The rationality of this modeling can be guaranteed by the Prop. C.1, App. C.1. Due to the linear evolution in the Koopman space, the mapping between latent states at time $s$ and $t$ for level $l$ and location $(i,j)$ can be explicitly expressed as

$$\boldsymbol{z}_t^{(l)}(i,j) = e^{\boldsymbol{A}^{(l)}(i,j)(t-s)} \boldsymbol{z}_s^{(l)}(i,j). \tag{5}$$

The explicit formulation enables the network to learn sufficient one-step mappings from $\boldsymbol{z}_t, \forall t \in [\epsilon, T]$ to $\boldsymbol{z}_\epsilon$ during the training process. Consequently, it allows direct mapping from $\boldsymbol{w}_T$ to $\boldsymbol{w}_\epsilon$ in the inference time, thus achieving efficient one-step sampling without the need for iterative integration.

**Decoder.** Finally, at the target time step $\epsilon$, the evolved set of evolved latent features $\{\boldsymbol{z}_\epsilon^{(l)}\}_{l=1}^L$ is decoded to generate the sample $\boldsymbol{x}_\epsilon$. The decoding process is performed by a decoder $\mathcal{D}\phi$, which implements the mapping

$$\mathcal{D}_\phi : \{\boldsymbol{z}_\epsilon^{(l)}\}_{l=1}^L \mapsto \boldsymbol{x}_\epsilon \in \mathbb{R}^{C \times H \times W}.$$

At each decoding level, the decoder upsamples the latent features from the lower-resolution level and integrates them with skip features obtained by evolving the encoder outputs through Koopman dynamics, just as the left part of Fig. 1.

## 3.2 Intermediate State Supervision: Trajectory Consistency Loss

A key advantage of our explicit framework over implicit mappings is the ability to supervise intermediate states during generation. Unlike conventional methods that supervise only the endpoint, our approach explicitly constrains the trajectory from noise to image by enforcing consistency with

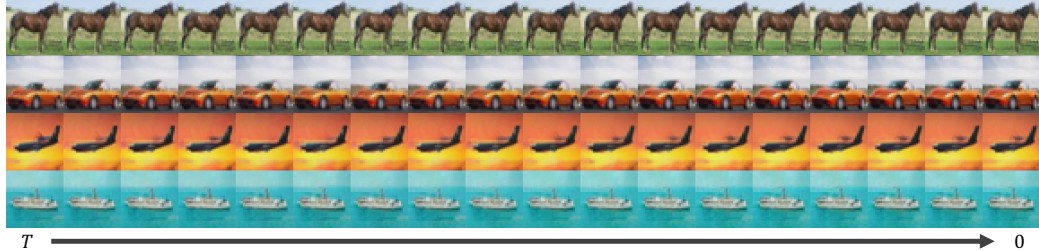

$T$ ⟶ $0$

Figure 2: Image reconstruction from noisy inputs at time $t$ highlights the consistency enforced by our framework. The left-most image is obtained from $\boldsymbol{x}_T$ and the right-most is from $\boldsymbol{x}_0$.

expected denoising dynamics along the path. Hence, we introduce a new trajectory consistency loss for the supervision of intermediate states. Fig. 2 presents the reconstructed images from noisy inputs at all time steps, showcasing the consistency properties promoted by our proposed loss function.

**Definition 3.1** (Trajectory Consistency Loss). *The trajectory consistency loss is defined as*

$$\mathcal{L}_{t\text{-consist}} \triangleq \mathbb{E}_{t\sim\mathcal{U}[\epsilon,T]}\Big[d\Big(\mathcal{D}_{\boldsymbol{\phi}}[\{e^{(\epsilon-t)\boldsymbol{A}^{(l)}}\mathcal{E}_{\boldsymbol{\theta}}^{(l)}(\boldsymbol{x}_t)\}_{l=1}^L], \boldsymbol{x}_\epsilon\Big)\Big], \tag{6}$$

*where the expectation is taken with respect to $t\sim\mathcal{U}[\epsilon,T]$, and $\boldsymbol{x}_\epsilon$ is given by Eq. (1). Distance $d(\cdot,\cdot)\geq 0$ is in the image space and equals 0 if and only if the argument variables are the same.*

This trajectory consistency loss enforces that any intermediate state $\boldsymbol{x}_t$, after being encoded and evolved through Koopman dynamics to time $\epsilon$, yields a decoded prediction that matches the ground truth image $\boldsymbol{x}_\epsilon$ obtained by Eq. (1). However, the most direct supervision would be to enforce alignment between the encoder-derived Koopman representation at time $t$ and its analytically evolved counterpart from time $T$ within the Koopman space. Although conceptually straightforward, enforcing consistency directly in the latent space may lead to suboptimal results in practice, as it reduces training flexibility and introduces gradients that poorly reflect perceptual discrepancies. Therefore, we instead adopt an alternative image-space formulation in Eq. (6), which enables the use of perceptually meaningful distance metrics, such as the Learned Perceptual Image Patch Similarity (LPIPS), to better capture semantic fidelity. Importantly, we show in App. C.2 that, under structural assumptions, minimizing the trajectory consistency loss in image space is theoretically equivalent to minimizing its latent-space counterpart.

### 3.3 Implementation

**Training.** We train the model using a total loss $\mathcal{L} = \mathcal{L}_{t\text{-consist}} + \mathcal{L}_{\text{recon}}$, where $\mathcal{L}_{t\text{-consist}}$ enforces consistency along the diffusion trajectory, and $\mathcal{L}_{\text{recon}} = d\Big(\mathcal{D}_{\boldsymbol{\psi}}(\{e^{(\epsilon-T)\boldsymbol{A}^{(l)}}\mathcal{E}_{\boldsymbol{\theta}}^{(l)}(\boldsymbol{x}_T)\}_{l=1}^L), \boldsymbol{x}_\epsilon\Big)$ supervises accurate one-step mapping from noise to clean image. The distance function is defined as $d(\boldsymbol{x}, \boldsymbol{y}) = \lambda_1\mathcal{L}_{\text{MSE}} + \lambda_2\mathcal{L}_{\text{LPIPS}}$, with $\lambda_2 = 1$ and $\lambda_1$ annealed to shift from coarse alignment to perceptual refinement. The full model, $\mathcal{E}_{\boldsymbol{\theta}}$, $\mathcal{D}_{\boldsymbol{\phi}}$, and $\{\boldsymbol{A}^{(l)}\}_{l=1}^L$, is trained end-to-end. The training algorithm is presented in Alg. 1. More training details are provided in App. B.1.

**One-Step Sampling.** At inference time, we could obtain the final generated sample $\hat{\boldsymbol{x}}_\epsilon$ from noise $\boldsymbol{x}_T$ by

$$\mathcal{D}_{\boldsymbol{\phi}}(\{e^{(\epsilon-T)\boldsymbol{A}^{(l)}}\mathcal{E}_{\boldsymbol{\theta}}^{(l)}(\boldsymbol{x}_T)\}_{l=1}^L).$$

Specifically, Given a noise sample, we project it into the Koopman space, perform one-step evolution via Koopman dynamics, and decode the result back to the data space. The detailed sampling algorithm of sampling is presented in Alg. 2.

**Koopman Spectral Analysis.** Our framework offers a distinct perspective on diffusion generation through Koopman spectral analysis. Each block $\boldsymbol{\Lambda}_k^{(l)}(i,j)$ in Eq. (4) represents a local dynamical mode, with eigenvalues $\alpha_k^{(l)}(i,j) \pm i\beta_k^{(l)}(i,j)$ encoding the growth/decay and oscillation of the trajectory. This explicit spectral interpretability further allows for controllable image generation by selectively modifying certain frequency components. More details of Koopman spectral analysis of our framework can be found in App. B.2.

---

**Algorithm 1** The training algorithm.

---

1: **Inputs:** Trajectory $\{\boldsymbol{x}_t\}_{t\in[0,T]}$ from a well-trained diffusion model; The number of samples $s$
2: **Outputs:** Network parameters $\boldsymbol{\theta}$ and $\boldsymbol{\phi}$; Koopman matrices $\{\boldsymbol{A}^{(l)}\}_{l=1}^{L}$
3: Initialize $\boldsymbol{\theta}$ and $\boldsymbol{\phi}$ by a pre-trained U-Net
4: Initialize $\boldsymbol{A}^{(l)} \leftarrow \boldsymbol{O}$, $l = 1, \cdots, L$
5: **for** $i = 0$ **to** num_iter$-1$ **do**
6:      Sample $S_t = \{t_i \mid t_i \sim \mathcal{U}[0,T]\}_{i=1}^{s-1} \cup \{T\}$                $\triangleright$ Uniformly sample the intermediate time
7:      Reconstruct $\hat{\boldsymbol{x}}_\varepsilon = \mathcal{D}_{\boldsymbol{\phi}}(\{\mathrm{e}^{(\epsilon-t)\boldsymbol{A}^{(l)}}\mathcal{E}_{\boldsymbol{\theta}}^{(l)}(\boldsymbol{x}_t)\}_{l=1}^{L})$ for $t \in S_t$      $\triangleright$ Apply the HKD model
8:      Let $\mathcal{L} = \sum_{t\in S_t} \left[ \|\hat{\boldsymbol{x}}_\varepsilon - \boldsymbol{x}_0\| + \|\mathcal{F}(\hat{\boldsymbol{x}}_\varepsilon) - \mathcal{F}(\boldsymbol{x}_0)\| \right]$      $\triangleright$ $\mathcal{F}$ is the feature extractor in LPIPS
9:      Update $\boldsymbol{\theta}$, $\boldsymbol{\phi}$ and $\boldsymbol{A}^{(l)}$ by the gradients of $\mathcal{L}$
10: **end for**
11: **Return** $\boldsymbol{\theta}$, $\boldsymbol{\phi}$ and $\{\boldsymbol{A}^{(l)}\}_{l=1}^{L}$

---

---

**Algorithm 2** The sampling algorithm.

---

1: **Inputs:** Trained HKD including network parameters $\boldsymbol{\theta}$, $\boldsymbol{\phi}$ and Koopman matrices $\{\boldsymbol{A}^{(l)}\}_{l=1}^{L}$
2: **Outputs:** Predicted image $\hat{\boldsymbol{x}}_\varepsilon = \mathcal{D}_{\boldsymbol{\phi}}(\{\mathrm{e}^{(\epsilon-T)\boldsymbol{A}^{(l)}}\mathcal{E}_{\boldsymbol{\theta}}^{(l)}(\boldsymbol{x}_T)\}_{l=1}^{L})$

---

## 3.4 Theoretical Results

In order to illustrate the feasibility of our proposed method, we theoretically analyze the error bounds caused by the proposed method, namely $err_{\mathrm{HKD}}$, and by conventional one-step methods $err_{\text{one-step}}$. Intuitively, neural networks of the same size are faced with larger errors when estimating more complex functions. In the proposed formulation, the encoder and decoder are commonly simple due to the elementary observable function, hence, the HKD generation, serving as a composition of three functions, is overall simpler than end-to-end one-step methods. Algebraically speaking, we have $err_{\mathrm{HKD}} \leq err_{\text{one-step}} + O(\kappa)$ where $\kappa$ is a reasonably small error caused by the Koopman process.

This inequality holds mainly because the proposed framework first transfers the complicated data space to the simpler Koopman space at a small cost and then controls the Koopman trajectory error by the explicit formulation in the Koopman theory. Thm. 3.1 provides an informal description of the conclusion where $\kappa = O(N^{-\frac{1}{2}}) + o(m^{-\frac{r}{3}})$ is small and that substantiates the common knowledge with $N$ and $m$ being the size of the dataset and the dimension of the Koopman space respectively. A formal description and proof of the theorem are given in App. C.3.

**Theorem 3.1** (Comparison of Error Bounds for One-Step Diffusions). *Let $err_{HKD}$ and $err_{one\text{-}step}$ be the ideal estimation errors of our proposed HKD model and an end-to-end one-step model with the same total number of activation functions we have,*

$$err_{HKD} \leq err_{one\text{-}step} + O(\kappa),$$

*where* $\kappa = \max\left\{ \dfrac{2\sqrt{3}\sigma m^2 \rho_{\sup}}{\sqrt{N(\frac{\delta}{3} - 2m^{-\frac{r}{3}})} - \sqrt{3}\sigma m^2}, \dfrac{2\sqrt{3}\sigma m^2 \rho_{\inf}}{\sqrt{N(\frac{\delta}{3} - 2m^{-\frac{r}{3}}) \cdot \rho_{\inf}} - \sqrt{3}\sigma m^2} \right\} + o(m^{-\frac{r}{3}})$ *with $N$ and $m$ dominating $\kappa$ being the size of the dataset and the dimension of the Koopman space respectively. The parameter $\sigma$ is the standard deviation of the observables, $\delta$ is $0$ when the network is perfectly trained, $\rho_{\inf}$ and $\rho_{\sup}$ are the true spectral range of the Koopman operator, and $r \to -\infty$ when the $m$-dimensional Koopman process adequately model the Koopman space.*

To formally prove the above result, we first proposed to evaluate the network size by the number of activation functions it uses, as they are the ones causing non-linearity of the network. Secondly, we introduced the simplicial error as the deviation in the simplicial complex estimation. Fig. 3 empirically shows that the metric of same-density triangulations reveals the complexity of 2D surfaces, demonstrating our idea. Consequently, the simplicial error quantitatively evaluates the complexity of the data and noise spaces and the mappings between them. Finally, using the ideas above, we analytically prove Thm. 3.1.

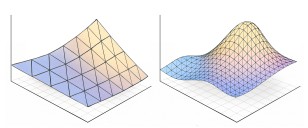

Figure 3: More complex surfaces need denser triangulation for the same error.

# 4 Experiments

We conducted experiments to present the promising one-step generation capability of our framework compared with other one-step baseline methods (Sec. 4.1). Notably, beyond generating high-quality images, we further provide empirical insights toward understanding the underlying dynamics of diffusion models through Koopman spectral analysis (Sec. 4.2). This analysis demonstrates the dynamical interpretability of our approach and further enables controllable image editing (Sec. 4.3). An ablation study was finally performed to show the key contributions of our framework (Sec. 4.4). Extensive results highlight both the generation quality and interpretability advantages of our framework, distinguishing it from distillation- and consistency-model-based paradigms.

## 4.1 Compare with Prior One-step Generation Methods

In this section, we evaluated our proposed framework on the CIFAR-10 [12] and FFHQ datasets [9], focusing exclusively on one-step generation methods within the diffusion model family. We used the standard Fréchet Inception Distance (FID) [6] metric and report FID-50k scores, following prior work [42]. Our model employs the U-Net [26] architecture from EDM [8] as the backbone for both the encoder and decoder, initialized with pretrained weights. All models were trained using the Adam optimizer [11] with a constant learning rate of $1 \times 10^{-3}$ and a weight decay of 0.95. App. D.4 provides additional implementation details of the experiments.

Table 1: Sample quality on CIFAR-10 dataset.

| Methods | NFE($\downarrow$) | FID($\downarrow$) |
|---|---|---|
| **Multi-Step Diffusion Models** | | |
| DDPM [7] | 1000 | 3.17 |
| Score SDE [33] | 2000 | 2.38 |
| DDIM [30] | 100 | 4.16 |
| DDIM [30] | 10 | 13.36 |
| EDM [8] | 35 | 1.97 |
| EDM [8] | 15 | 5.62 |
| **Diffusion Distillation** | | |
| KD [17] | 1 | 9.36 |
| PD [27] | 1 | 8.34 |
| CD (LPIPS) [32] | 1 | 3.55 |
| DMD [39] | 1 | 3.77 |
| 1-Rectified flow [14] | 1 | 6.18 |
| 2-Rectified flow [14] | 1 | 4.85 |
| 3-Rectified flow [14] | 1 | 5.21 |
| 2-Rectified flow++ [13] | 1 | 3.38 |
| **Consistency Model** | | |
| CT (LPIPS) [32] | 1 | 8.70 |
| CD (LPIPS) [32] | 1 | 3.55 |
| iCT [31] | 1 | 2.83 |
| iCT-deep [31] | 1 | 2.51 |
| ECM [4] | 1 | 3.60 |
| HKD | 1 | 3.30 |

**Comparison to Prior Work.** We evaluated our framework against two dominant one-step generation approaches: distillation-based methods and consistency models, with comparisons to multi-step diffusion baselines for context. The multi-step diffusion baselines include: (1) **DDPM** [7]: The foundational denoising diffusion model; (2) **Score SDE** [33]: Continuous-time diffusion via reverse-time SDEs; (3) **DDIM** [30]: Efficient deterministic sampler from diffusion ODEs; (4) **EDM** [8]: State-of-the-art diffusion backbone (used in our framework).

For the distillation-based one-step paradigm, the comparison extends to traditional leading methods: (1) **KD** [17]: Traditional knowledge distillation method with direct teacher-student mimicry; (2) **PD** [27]: Progressive distillation with step reduction through sequential student-teacher training; (3) **CD (LPIPS)** [32]: Consistency distillation with LPIPS loss; (4) **DMD** [39]: Distribution matching distillation with the distribution-matching loss; (5) and **(distilled) ReFlow** series: including 1-Rectified flow, 2-Rectified flow, 3-Rectified flow [14] and 2-Rectified flow++ [13] with reflow and distillation. For the consistency-model-based method, we included: (1) **CT** [32]: Consistency training including CT (LPIPS), iCT [31], and iCT-deep [31], which provides a comparison to the end-to-end traditional consistency model, and its improved versions. (2) **ECM** [4]: Easy Consistency Models, which initializes the network weights with the ones from a pretrained score model. More details of comparison methods are in App. D.5.

**Results.** While enhanced methods such as iCT-deep [31] have pushed performance boundaries, they suffer from high sensitivity to hyperparameters. Moreover, state-of-the-art consistency training typically requires nearly a week of training on 8 GPUs [4], and remains unstable in practice [16]. Retraining with consistency distillation (CD) even results in a degraded FID of 10.53, as reported

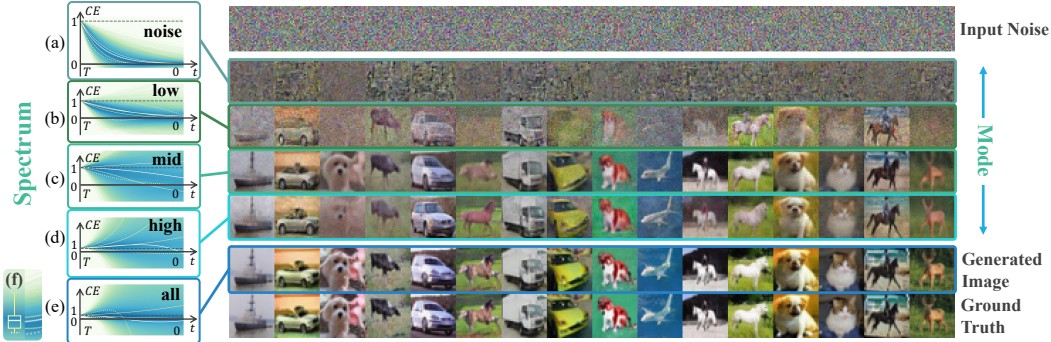

Figure 4: Visualization of Spectral Contributions in Generated Images Across Koopman Modes. (a) visualizes the contribution of Koopman spectrum with the smallest real-part magnitudes to the reconstructed state over time. It corresponds to the noisy part of the image. (b), (c) and (d) illustrate progressively larger spectral components and their reconstructed image components.

in [10]. In contrast, our method achieved comparable performance within just 2–3 days on 8×V100 GPUs.

Moreover, we significantly improved the training stability thanks to two design choices: (1) the exponential formulation in the Koopman space ensures sufficiently large gradients for spectra with magnitudes near 1, mitigating the high-variance gradient issues commonly seen in consistency models [28]; and (2) supervising the Koopman trajectory at multiple time points enables more stable and averaged spectrum estimation, leading to more reliable mode evolution modeling. Experimental evidence of the improved training stability on the CIFAR-10 dataset is provided in App. D.1.

Table 2: Sample quality on FFHQ.

| Methods | NFE($\downarrow$) | FID($\downarrow$) |
|---------|------|------|
| DDIM [30] | 10 | 18.30 |
| EDM [8] | 79 | 2.47 |
| EDM [8] | 15 | 9.85 |
| ECM [4] | 1 | 5.99 |
| HKD | 1 | 5.70 |

Additional experiments on FFHQ 64x64 (Tab. 2) validated the effectiveness of our method on the high-resolution and structurally complex dataset. See App. D.1 for additional visualizations, per-image wall-clock timings for various generation methods, and conditional generation results.

## 4.2 Koopman Spectral Analysis for Generative Process

We conducted an analysis of $A$ to further investigate how individual spectral components influence the generative trajectory. We first tracked the contribution of each Koopman spectrum (cumulative effect, namely CE) to the reconstruction over time. Then, at each resolution level $l$, we sorted the eigenvalue pairs across spatial locations by the real parts and applied spectral masking by retaining only those within a target range (smallest, intermediate, or largest), zeroing out the rest. The masked representations are decoded to visualize the contribution of selected spectral modes.

Fig. 4 summarizes the results of the two spectral analyses: the left shows the contribution of grouped Koopman modes to reconstruction over time, while the right links spectral ranges to corresponding image structures. By selectively activating spectral bands and decoding the latent representations, we observe a clear semantic hierarchy: low-range modes capture global structure, mid-range modes recover overall shape and pose, and high-range modes refine local details. These results demonstrate the semantic interpretability of Koopman spectra and their potential for controllable image synthesis. More Koopman spectral analysis results are provided in App. D.2.

## 4.3 One-Step Image Editing: A Case of Model Interpretability

We evaluated the interpretability of our framework through an image editing experiment that targets frequency-specific interventions at the intermediate state of the diffusion trajectory. We performed editing by injecting high-frequency features from a reference image into the generated image at mixing ratios of 10%, 20%, 50%, 80%, and 90%, applied at the middle time step to demonstrate

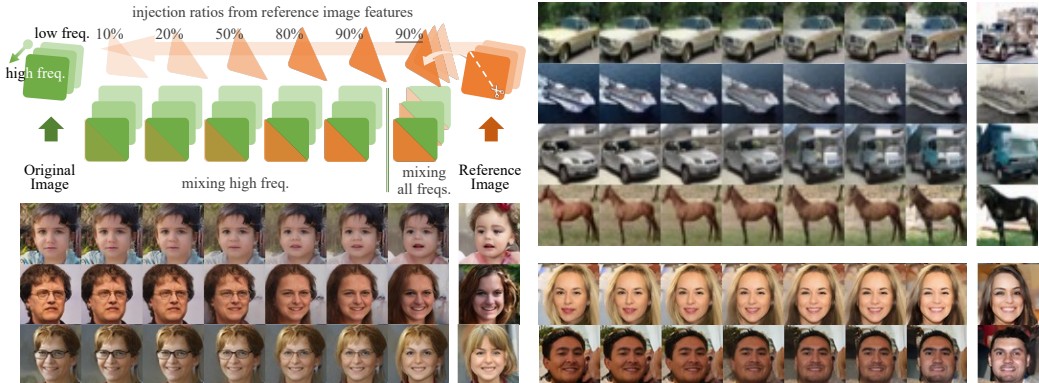

Figure 5: One-step image editing via frequency-aware interventions along the diffusion trajectory. We controlled the image generation by the high-frequency features from a reference image through injecting them into the lower-left half of the generating image at different mixing ratios (10%, 20%, 50%, 80%, 90%). The modifications were performed at the midpoint of the Koopman trajectory. Columns 2-6 showcase the frequency-aware editing, where only high-frequency components were mixed, preserving the low-frequency structure of the original image. Column 7 is for frequency-agnostic editing, where all-frequency features of reference images are mixed with all frequency bands of the original image at a mixing ratio of 90%. We exhibit the results from both datasets.

temporal editability. For comparison, frequency-agnostic editing mixes the full-spectrum content of the reference and generated images at the same step.

As shown in Fig. 5, increasing the injection ratio of reference image features from 10% to 90% gradually reveals more facial details from the reference, demonstrating that our frequency-aware editing establishes meaningful correspondences through interpretable frequency decomposition. In contrast, frequency-agnostic editing disrupts global structures, indicating a lack of disentangled control. These results further validate the interpretability of our framework.

We further perform an image editing experiment of image recuperation in the impainting and coloring tasks on CIFAR-10 dataset following Algorithm 4 from [32], which iteratively mixes a reference image with the generated image along the generative trajectory by adding and removing noise at each time step $t$. The results are presented in Fig. 6.

## 4.4 Ablation Study

We present an ablation study on the CIFAR-10 dataset to evaluate the contributions of key components in our framework: Koopman evolution (Koop.), the trajectory consistency loss ($\mathcal{L}_{t\text{-consist}}$), and hierarchical design (hierar.). Tab. 3 shows that the one-step baseline model removing the Koopman dynamics results in an FID of 5.72. Upon it, additional Koopman evolutions at the skips and bottleneck paths, and introducing trajectory consistency loss further improve the results. In addition, a hierarchical Koopman dynamics design also contributes to better generation quality.

Table 3: Ablation study on CIFAR-10.

| Settings | | | FID ($\downarrow$) |
|---|---|---|---|
| Koop. | $\mathcal{L}_{t\text{-consist}}$ | hierar. | |
| ✗ | ✗ | ✓ | 5.72 |
| ✓ | ✗ | ✓ | 5.57 |
| ✓ | ✓ | ✗ | 4.78 |
| ✓ | ✓ | ✓ | 3.30 |

## 5 Related Work

**One-step Generation for Diffusion Models.** Prior works have distilled pretrained diffusion models into efficient one-step generators. Knowledge distillation [17] trains a student model to replicate the denoising behavior of the teacher, rectified flows [14] reformulate diffusion as ODEs for distillation, all considered as classical approaches. Recent works further improve distillation-based generation, such as DMD [39], which aligns one-step generators with diffusion models via KL divergence, and SlimFlow, which [42] enhances rectified flows through compression of model size. Consistency

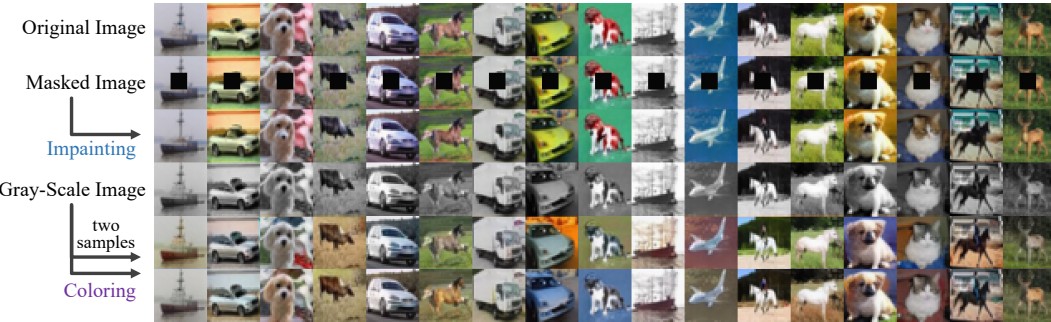

Figure 6: The visualization of image recuperation in the impainting and coloring tasks. The experiment is performed by Algorithm 4 from [32], which iteratively mixes the image with a reference image created by adding and removing noise at time $t$ with $t$ decreasing from $T$ to $0$.

models [32] take a different route, suffering from instability and inefficiency. Improvements include fine-tuning from diffusion models [4] and adversarial enhancements [5]. However, these methods prioritize performance at the cost of losing the inherent interpretability and controllability offered by diffusion models.

**Koopman Operators.** Koopman operator theory has enabled advances in system analysis [19], control [23], and optimization [25] by extracting globally linear structures from nonlinear dynamics. Recent work extends its use to time series modeling [22, 40, 2]. In addition, many other works have proposed approximating the Koopman operators using neural networks. For instance, KNF [35] leverages DNNs to learn the linear Koopman space and the coefficients of chosen measurement functions. To our knowledge, we are the first to introduce Koopman theory into the field of image generation. See App. A.1 for more related work.

## 6 Discussion

HKD is an interpretable one-step generation framework that retains access to intermediate states and fine-grained control—while enabling one-step sampling. It is the first to introduce interpretability and intermediate-state intervention into one-step generation. By bridging fast sampling and interpretability, HKD opens new possibilities for explainable image synthesis. While our current formulation already demonstrates competitive one-step generation performance and enables spectral interventions, several directions remain open for future exploration. First, although we adopt a standard training paradigm without relying on adversarial optimization or heavy tuning, integrating advanced training techniques (e.g., adversarial learning) may further improve generation quality. In addition, while HKD performs well on standard-resolution datasets, its potential for high-resolution generation, afforded by its hierarchical design, remains underexplored. Moreover, the explicit spectral decomposition in our framework naturally supports interpretable interventions, making it well-suited for a range of semantic editing tasks. Beyond the frequency-aware manipulations we demonstrate, HKD opens up possibilities for text-guided editing, attribute-specific control. While these directions are currently unexplored, they underscore the broader applicability of our framework beyond pure generation. Our framework provides a solid foundation for these future advancements.

## Acknowledgments

We would like to thank the anonymous reviewers and area chairs for their helpful comments. This work was partially supported by the National Natural Science Foundation of China (No. 12471481, U24A2001), the Science and Technology Commission of Shanghai Municipality (No. 23ZR1403000), and the Open Foundation of Key Laboratory Advanced Manufacturing for Optical Systems, CAS (No. KLMSKF202403). We also acknowledge the support (to D. Zou) from NSFC 62306252, Hong Kong ECS award 27309624, Guangdong NSF 2024A1515012444, and the central fund from HKU IDS.

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

## Appendix Overview

The following lists the structure of the appendix, with links to the respective sections.

# A  Background Knowledge Supplement

## A.1   Related Work

**One-step Generation for Diffusion Models.**   Several prior works have explored distilling pretrained diffusion models into efficient one-step generators. Knowledge distillation [17] trains a student model to mimic the full sampling trajectory of a pretrained diffusion model. Rectified flows [14] reformulate diffusion sampling as a continuous normalizing flow, allowing distillation via ODE simulation. Progressive distillation [27] gradually reduces the number of inference steps during training, enabling faster sampling with minimal performance loss. Recent works further improve distillation-based generation, such as SiD [41], which reformulates forward diffusion processes as semi-implicit distributions, DMD [39], which enforces the one-step image generator to match the diffusion model at the distribution level by minimizing an approximate KL divergence, and SIM [18], which computes the gradients for a wide class of score-based divergences between a diffusion model and a generator. In parallel, SlimFlow [42] builds on rectified flows by exploring joint compression of inference steps and model size to enhance efficiency. Consistency models [32] offer an alternative approach, learning to map noisy inputs directly to clean outputs. These models can either be trained end-to-end or used as distillation students. Due to their training instability and inefficiency, various improvements have been proposed, such as [4], which fine-tunes a consistency model starting from a pretrained diffusion model, [28], which trains consistency models with variational noise coupling, and [5] by enhancing consistency models via adversarialy-trained classification and energy-based discrimination. However, most of these works ignore the interpretability of the generation process.

**Koopman Operators.**   Over the past two decades, Koopman operator theory has attracted growing interest, enabling progress in dynamical system analysis [36, 38, 37], control [23], optimization [25], and forecasting [15]. These methods utilize Koopman-based representations to extract globally linearizable structures from nonlinear dynamics, improving interpretability and control. More recently, Koopman operators have been increasingly applied to time series modeling [22, 40]. In addition, several works have leveraged neural networks to approximate Koopman operators. For example, [1] proposes balanced neural ODEs to approximate Koopman operators without predefined dimensionality, while [**?** ] uses DNNs to learn both the Koopman space and coefficients of selected observables. In our work, we similarly employ neural networks to learn observable functions for end-to-end training. To the best of our knowledge, we are the first to introduce Koopman theory into image generation.

## A.2   Koopman Operator: From Infinite-Dimensional to Finite-Dimensional Approximation

In theory, the Koopman operator $\mathcal{K}$ is an infinite-dimensional linear operator that governs the evolution of observable functions under a nonlinear dynamical system. That is, for a dynamical map $\Phi : \mathcal{X} \to \mathcal{X}$ and any observable function $g : \mathcal{X} \to \mathbb{R}$, the Koopman operator acts as

$$(\mathcal{K}g)(\boldsymbol{x}) = g(\Phi(\boldsymbol{x})). \tag{7}$$

However, this operator acts on an infinite-dimensional function space, which is not directly tractable in practice. To enable computation, the Koopman operator is commonly approximated in a finite-dimensional subspace. This is achieved by selecting a finite set of basis functions $\boldsymbol{u} = [u_1, u_2, \ldots, u_m]^\top$ to represent the space of observables. We then approximate the action of $\mathcal{K}$ on this basis as

$$\mathcal{K}\boldsymbol{u}(\boldsymbol{x}) \approx \boldsymbol{K}\boldsymbol{u}(\boldsymbol{x}), \tag{8}$$

where $\boldsymbol{K} \in \mathbb{R}^{m \times m}$ is a finite-dimensional matrix known as the Koopman matrix. Given an observable $g$ projected onto the subspace as $\boldsymbol{\underline{\xi}} \triangleq [\langle g, u_1\rangle, \langle g, u_2\rangle, \cdots, \langle g, u_m\rangle]^\top$, its evolution under $\Phi$ can be approximated by $g(\Phi(\boldsymbol{x})) \approx \boldsymbol{\xi}^\top \boldsymbol{K}\boldsymbol{u}(\boldsymbol{x})$. Furthermore, if we consider a vector of observables $\boldsymbol{g} \approx \boldsymbol{P}\boldsymbol{u}$ that is linearly related to the basis via some invertible matrix $\boldsymbol{P}$, the Koopman dynamics in this new coordinate system are given by the similarity transform:

$$\boldsymbol{g}(\Phi(\boldsymbol{x})) \approx \boldsymbol{P}\boldsymbol{K}\boldsymbol{P}^{-1}\boldsymbol{g}(\boldsymbol{x}). \tag{9}$$

Several numerical algorithms have been developed to compute the Koopman matrix $\boldsymbol{K}$ from data:

(1) Dynamic Mode Decomposition (DMD): DMD is a widely used data-driven technique that approximates the Koopman operator using linear regression on snapshots of system states. It assumes

a linear relationship between time-shifted observables and solves for the best-fit linear operator $\boldsymbol{K}$ such that $\boldsymbol{x}_{t+1} \approx \boldsymbol{K}\boldsymbol{x}_t$ in the observable space. Variants include Extended DMD (EDMD) and Hankel-DMD for richer function spaces.

(2) Extended Dynamic Mode Decomposition (EDMD): EDMD generalizes DMD by applying the method in a lifted feature space defined by nonlinear basis functions (e.g., polynomials, radial basis functions). This corresponds to choosing a specific $\boldsymbol{u}$ and solving for the best Koopman matrix in this subspace.

(3) Neural Approximations: More recently, neural networks have been used to learn either the Koopman embedding (i.e., the basis functions $\boldsymbol{u}$) or the Neural Approximations use trainable architectures to discover a latent space where linear evolution via a Koopman operator holds approximately. They enable scalable and adaptive modeling of complex, high-dimensional dynamics. In our work, we adopt this method to approximate the Koopman operators.

# B  Framework Supplement

## B.1  Training Details

We optimize our model using a total loss that combines the trajectory consistency loss $\mathcal{L}_{t\text{-consist}}$ that supervises intermediate states along the diffusion trajectory with a reconstruction loss

$$\mathcal{L}_{\text{recon}} = d\Big(\mathcal{D}_{\boldsymbol{\psi}}(\{\mathrm{e}^{(\epsilon-T)\boldsymbol{A}^{(l)}}\mathcal{E}_{\boldsymbol{\theta}}^{(l)}(\boldsymbol{x}_T)\}_{l=1}^{L}), \boldsymbol{x}_{\epsilon}\Big)$$

from the noise $\boldsymbol{x}_T$ to enforce the accurate one-step mapping from noise to clean image, i.e.,

$$\mathcal{L} = L_{t\text{-consist}} + L_{\text{recon}}.$$

Define that $\boldsymbol{x}_{\epsilon}^t \triangleq \mathcal{D}_{\boldsymbol{\psi}}(\{\mathrm{e}^{(\epsilon-t)\boldsymbol{A}^{(l)}}\mathcal{E}_{\boldsymbol{\theta}}^{(l)}(\boldsymbol{x}_t)\}_{l=1}^{L})$, then $\mathcal{L}_{\text{total}} = \sum_{t \in \{T\} \cup \mathcal{T}} d(\boldsymbol{x}_{\epsilon}^t, \boldsymbol{x}_{\epsilon})$, where the elements of $\mathcal{T}$ are sampled from $U[\epsilon, T]$ and $d(\boldsymbol{x}, \boldsymbol{y})$ is a metric composed of $l_2$-distance, i.e., MSE loss, and LPIPS loss in our model:

$$d(\boldsymbol{x}, \boldsymbol{y}) = \lambda_1 \mathcal{L}_{\text{MSE}}(\boldsymbol{x}, \boldsymbol{y}) + \lambda_2 \mathcal{L}_{\text{LPIPS}}(\boldsymbol{x}, \boldsymbol{y}).$$

To balance perceptual fidelity and optimization stability, we fix $\lambda_2 = 1$ and apply an annealing schedule to $\lambda_1$, allowing the model to initially focus on coarse alignment before gradually emphasizing perceptual refinement.

For network architecture, we adopt the mature U-Net encoder and decoder backbone used in diffusion models for both the encoder and decoder modules in our formulation to leverage existing diffusion models' design efficacy. To improve training efficiency and reduce mode collapse, we further initialize the encoder and decoder with pre-trained weights from diffusion models, which provide structured latent-to-output mappings, and retain rich hierarchical features acquired during large-scale training. The entire model, including $\mathcal{E}_{\boldsymbol{\theta}}$, $\mathcal{D}_{\boldsymbol{\phi}}$, and $\{\boldsymbol{A}^{(l)}\}_{l=1}^{L}$, is trained end-to-end using the loss $\mathcal{L}_{\text{total}}$.

## B.2  Koopman Spectral Analysis: A New Lens on Interpreting Diffusion Dynamics

Our framework offers a fundamentally different viewpoint on the analysis of the diffusion generation process via Koopman spectral analysis. Beyond enabling one-step sampling, the proposed formulation naturally provides a novel dynamical perspective of diffusion-based generative models through the lens of Koopman spectral analysis. By explicitly modeling the latent evolution as a linear dynamical system, our approach allows for the application of spectral tools from dynamical systems theory to reveal the underlying spectral structure of generative dynamics and provide new theoretical tools for analyzing and interpreting dynamical modes during sampling.

Specifically, each block $\boldsymbol{\Lambda}_k^{(l)}(i, j)$ directly characterizes a local dynamical mode, with eigenvalues $\alpha_k^{(l)}(i, j) \pm i\beta_k^{(l)}(i, j)$ encoding the growth/decay rate and oscillatory frequency of the diffusion trajectories. The eigenvalues of Koopman matrix $\boldsymbol{K}^{(l)}(i, j)$ associated with each mode can be computed as

$$\lambda_k\Big(\boldsymbol{K}^{(l)}(i, j)\Big) = e^{\alpha_k^{(l)}(i,j)\Delta t} \Big[\cos\Big(\beta_k^{(l)}(i, j)\Delta t\Big) \pm i\sin\Big(\beta_k^{(l)}(i, j)\Delta t\Big)\Big], \qquad (10)$$

providing a principled spectral description of the generative dynamics. The magnitude $|\lambda_k(\boldsymbol{K}^{(l)}(i,j)| = e^{\alpha_k^{(l)}(i,j)\Delta t}$ determines the stability of each mode, while the imaginary part of $\lambda_k(\boldsymbol{K}^{(l)}(i,j))$ captures the frequency modes of oscillations. Benefiting from the block-diagonal structure of $\boldsymbol{A}^{(l)}(i,j)$, our framework enables direct analysis of its spectral components to interpret the generative behavior of the model by directly observing the values of $\alpha_k^{(l)}(i,j)$ and $\beta_k^{(l)}(i,j)$. Specifically, the high real parts $\alpha_k^{(l)}(i,j)$ and imaginary parts $\beta_k^{(l)}(i,j)$ and low $\alpha_k^{(l)}(i,j)$ and $\beta_k^{(l)}(i,j)$ of each block correspond to low- and high-frequency modes in the generation process, respectively.

This explicit spectral interpretability further allows for controllable image editing by selectively modifying certain frequency components, enabling targeted modification of structural or fine-grained details in the generated images.

## C Theoretical Supplement

### C.1 Proposition: Rationality of the Block Diagonalization of $A$

We present the following proposition to ensure the rationality of the block diagonalization of the linear operator $\boldsymbol{A}$.

**Proposition C.1.** *We define the set for the l-layer encoders $S_{\mathcal{E}} \triangleq \{\mathcal{E} = \mathcal{E}_l \circ \cdots \circ \mathcal{E}_2 \circ \mathcal{E}_1 \mid \mathcal{E}_l : \mathfrak{f} \mapsto \boldsymbol{W}_l^{(\mathcal{E})}\mathfrak{f} + \boldsymbol{b}_l^{(\mathcal{E})}\}$ and the set for the decoders $S_{\mathcal{D}} \triangleq \{\mathcal{D} = \mathcal{D}_l \circ \cdots \circ \mathcal{D}_2 \circ \mathcal{D}_1 \mid \mathcal{D}_1 : \mathfrak{x} \mapsto \boldsymbol{W}_1^{(\mathcal{D})}\mathfrak{x} + \boldsymbol{b}_1^{(\mathcal{D})}\}$. Then for $\mathcal{E} \in S_{\mathcal{E}}, S_{\mathcal{D}} \in S_{\mathcal{D}}$ and the Koopman matrix $\boldsymbol{K}$, there exists another set of $\tilde{\mathcal{E}} \in S_{\mathcal{E}}, \tilde{\mathcal{D}} \in S_{\mathcal{D}}$ and $\tilde{\boldsymbol{K}} \in \mathbb{R}^{m \times m}$ that $\tilde{\boldsymbol{K}} = \boldsymbol{I} + diag\{\boldsymbol{\Lambda}_1, \boldsymbol{\Lambda}_2, \cdots, \boldsymbol{\Lambda}_k\}$ and $\boldsymbol{\Lambda}_i = \begin{bmatrix} \alpha_i & \beta_i \\ -\beta_i & \alpha_i \end{bmatrix}$, such that,*

$$\mathcal{D}e^{-t(\boldsymbol{K}-\boldsymbol{I})}\mathcal{E} \equiv \tilde{\mathcal{D}}e^{-t(\tilde{\boldsymbol{K}}-\boldsymbol{I})}\tilde{\mathcal{E}}. \tag{11}$$

*Proof.* We assume that the eigenvalue decomposition of matrix $\boldsymbol{K}$ in the complex field is $\boldsymbol{K} = \boldsymbol{P}\boldsymbol{\Lambda}\boldsymbol{P}^{-1}$ where $\boldsymbol{\Lambda} \triangleq \text{diag}\{\lambda_1, \bar{\lambda}_1 \cdots, \lambda_r, \bar{\lambda}_r, \lambda_{r+1}, \lambda_{r+2}, \cdots, \lambda_{m-r}\}$ with $\lambda_i, \bar{\lambda}_i, 1 \leq i \leq r$ the conjugated imaginary eigenvalues and $\lambda_i, r+1 \leq i \leq m-r$ the real eigenvalues.

Let $\boldsymbol{\Lambda}_i = \begin{bmatrix} \text{Re } \lambda_i - 1 & \text{Im } \lambda_i \\ -\text{Im } \lambda_i & \text{Re } \lambda_i - 1 \end{bmatrix}, 1 \leq i \leq r$ and $\boldsymbol{\Lambda}_i = \begin{bmatrix} \lambda_i - 1 & 0 \\ 0 & \lambda_i - 1 \end{bmatrix}, r+1 \leq i \leq m-r$, we use them to define $\tilde{\boldsymbol{K}}$. Furthermore, let $\tilde{\boldsymbol{P}} \triangleq [\text{Re } p_1, \text{Im } p_1, \cdots, \text{Re } p_r, \text{Im } p_r, \frac{p_{r+1}}{2}, \frac{p_{r+1}}{2}, \cdots, \frac{p_{m-r}}{2}, \frac{p_{m-r}}{2}]$ where $\boldsymbol{P} = [p_1, \bar{p}_1, \cdots, p_r, \bar{p}_r, p_{r+1}, p_{r+2}, \cdots, p_{m-r}]$ and $\tilde{\boldsymbol{Q}} = \sqrt{2} [\text{Re } p_1^{\dagger}, -\text{Im } p_1^{\dagger}, \cdots, \text{Re } p_r^{\dagger}, -\text{Im } p_r^{\dagger}, \frac{p_{r+1}^{\dagger}}{2}, \frac{p_{r+1}^{\dagger}}{2}, \cdots, \frac{p_{m-r}^{\dagger}}{2}, \frac{p_{m-r}^{\dagger}}{2}]^{\top}$ where $\boldsymbol{P}^{-1} = [p_1^{\dagger}, \bar{p}_1^{\dagger}, \cdots, p_r^{\dagger}, \bar{p}_r^{\dagger}, p_{r+1}^{\dagger}, p_{r+2}^{\dagger}, \cdots, p_{m-r}^{\dagger}]^{\top}$. Note that $p_i^{\dagger}$'s are row vectors and $\tilde{\boldsymbol{P}}\tilde{\boldsymbol{Q}}$ is NOT necessarily identity.

Last but not least, let $\tilde{\boldsymbol{W}}_l^{(\mathcal{E})} \triangleq \tilde{\boldsymbol{Q}}\boldsymbol{W}_l^{(\mathcal{E})}$ and $\tilde{\boldsymbol{b}}_l^{(\mathcal{E})} \triangleq \tilde{\boldsymbol{Q}}\boldsymbol{b}_l^{(\mathcal{E})}$ in $\tilde{\mathcal{E}}$ and $\tilde{\boldsymbol{W}}_1^{(\mathcal{D})} \triangleq \boldsymbol{W}_1^{(\mathcal{D})}\tilde{\boldsymbol{P}}$ in $\tilde{\mathcal{D}}$ then

$$\begin{aligned} &\tilde{\mathcal{D}}\text{e}^{-t(\tilde{\boldsymbol{K}}-\boldsymbol{I})}\tilde{\mathcal{E}}(\boldsymbol{x}), \\ =&\tilde{\mathcal{D}}_l \circ \cdots \circ \tilde{\mathcal{D}}_2(\tilde{\boldsymbol{W}}_1^{(\mathcal{D})}\text{e}^{-t(\tilde{\boldsymbol{K}}-\boldsymbol{I})}(\tilde{\boldsymbol{W}}_l^{(\mathcal{E})}\tilde{\mathcal{E}}_{l-1} \circ \cdots \circ \tilde{\mathcal{E}}_1(\boldsymbol{x}) + \tilde{\boldsymbol{b}}_l^{(\mathcal{E})}) + \tilde{\boldsymbol{b}}_1^{(\mathcal{D})}), \\ =&\mathcal{D}_l \circ \cdots \circ \mathcal{D}_2(\boldsymbol{W}_1^{(\mathcal{D})}\tilde{\boldsymbol{P}}\text{e}^{-t(\tilde{\boldsymbol{K}}-\boldsymbol{I})}\tilde{\boldsymbol{Q}}(\boldsymbol{W}_l^{(\mathcal{E})}\mathcal{E}_{l-1} \circ \cdots \circ \mathcal{E}_1(\boldsymbol{x}) + \boldsymbol{b}_l^{(\mathcal{E})}) + \boldsymbol{b}_1^{(\mathcal{D})}), \\ =&\mathcal{D}\left[\tilde{\boldsymbol{P}}\text{e}^{-t(\tilde{\boldsymbol{K}}-\boldsymbol{I})}\tilde{\boldsymbol{Q}}\ \mathcal{E}(\boldsymbol{x})\right], \\ \overset{\star}{=}&\mathcal{D}\text{e}^{-t(\boldsymbol{K}-\boldsymbol{I})}\mathcal{E}(\boldsymbol{x}), \end{aligned}$$

where $\overset{\star}{=}$ holds due to $\begin{bmatrix} \frac{p_i}{\sqrt{2}} & \frac{p_i}{\sqrt{2}} \end{bmatrix} \text{e}^{-t\boldsymbol{\Lambda}_i} \begin{bmatrix} p_i^{\dagger}/\sqrt{2} \\ p_i^{\dagger}/\sqrt{2} \end{bmatrix} = \text{e}^{-t(\lambda_i-1)}p_i p_i^{\dagger}, r+1 \leq i \leq m-r$ and

$$\left[ \sqrt{2}\mathrm{Re}\; p_i \quad \sqrt{2}\mathrm{Im}\; p_i \right] \mathrm{e}^{-t\boldsymbol{\Lambda}_i} \begin{bmatrix} \sqrt{2}\mathrm{Re}\; p_i^\dagger \\ -\sqrt{2}\mathrm{Im}\; p_i^\dagger \end{bmatrix}, \tag{12}$$

$$\overset{\lambda_i=\alpha_i+\mathrm{i}\beta_i}{=\!=\!=\!=\!=} 2 \left[ \mathrm{Re}\; p_i \quad \mathrm{Im}\; p_i \right] \begin{bmatrix} \mathrm{e}^{-t(\alpha_i-1)}\cos(-t\beta_i) & \mathrm{e}^{-t(\alpha_i-1)}\sin(-t\beta_i) \\ -\mathrm{e}^{-t(\alpha_i-1)}\sin(-t\beta_i) & \mathrm{e}^{-t(\alpha_i-1)}\cos(-t\beta_i) \end{bmatrix} \begin{bmatrix} \mathrm{Re}\; p_i^\dagger \\ -\mathrm{Im}\; p_i^\dagger \end{bmatrix}, \tag{13}$$

$$= 2\mathrm{e}^{-t(\alpha_i-1)} \left[ \mathrm{Re}\; p_i \quad \mathrm{Im}\; p_i \right] \begin{bmatrix} \cos(t\beta_i) & -\sin(t\beta_i) \\ \sin(t\beta_i) & \cos(t\beta_i) \end{bmatrix} \begin{bmatrix} \mathrm{Re}\; p_i^\dagger \\ -\mathrm{Im}\; p_i^\dagger \end{bmatrix}, \tag{14}$$

$$= \mathrm{e}^{-t(\alpha_i-1)} \left[ \mathrm{Re}\; p_i \quad \mathrm{Im}\; p_i \right] \begin{bmatrix} 1 & 1 \\ \mathrm{i} & -\mathrm{i} \end{bmatrix} \begin{bmatrix} \mathrm{e}^{-t\beta_i\mathrm{i}} & 0 \\ 0 & \mathrm{e}^{t\beta_i\mathrm{i}} \end{bmatrix} \begin{bmatrix} 1 & -\mathrm{i} \\ 1 & \mathrm{i} \end{bmatrix} \begin{bmatrix} \mathrm{Re}\; p_i^\dagger \\ -\mathrm{Im}\; p_i^\dagger \end{bmatrix}, \tag{15}$$

$$= \mathrm{e}^{-t(\alpha_i-1)} \left[ p_i \quad \bar{p}_i \right] \begin{bmatrix} \mathrm{e}^{-t\beta_i\mathrm{i}} & 0 \\ 0 & \mathrm{e}^{t\beta_i\mathrm{i}} \end{bmatrix} \begin{bmatrix} p_i^\dagger \\ \bar{p}_i^\dagger \end{bmatrix} = \left[ p_i \quad \bar{p}_i \right] \mathrm{e}^{-t\left( \begin{bmatrix} \lambda_i & 0 \\ 0 & \bar{\lambda}_i \end{bmatrix} - \boldsymbol{I} \right)} \begin{bmatrix} p_i^\dagger \\ \bar{p}_i^\dagger \end{bmatrix}, \tag{16}$$

for $1 \le i \le r$. $\qquad\square$

## C.2 Proposition: Image-Space Trajectory Consistency Loss and its Latent-Space Counterpart

In this part, we first propose the informal version of the equivalence between image-space trajectory consistency loss and its latent-space counterpart under specific assumptions. Then we present the formal version.

**Proposition C.2** (Equivalence of Trajectory Consistency Losses under Structural Assumptions, Informal). *Assume that the decoder $\mathcal{H} = \mathcal{F} \circ \mathcal{D}_\phi, \forall \mathcal{F}$ is differentiable with respect to each Koopman state $\boldsymbol{w}^{(l)}$, and satisfies isotropy and level disentanglement properties with respect to the Koopman decomposition. Then, the image-space trajectory consistency loss:*

$$\mathcal{L}_{t-consist} = \mathbb{E}_{t\sim\mathcal{U}[\epsilon,T]} \left[ d\left( \mathcal{D}_\phi(\{e^{(\varepsilon-t)\boldsymbol{A}^{(l)}} \mathcal{E}_{\boldsymbol{\theta}}^{(l)}(\boldsymbol{x}_t)\}_{l=1}^L), \boldsymbol{x}_\epsilon \right) \right], \tag{17}$$

*is approximately equivalent to the following latent-space loss:*

$$\mathcal{L}_{t-consist} \approx \mathbb{E}_{t\sim\mathcal{U}(\epsilon,T)} \sum_{l=1}^L \left\| \frac{\partial \mathcal{F}\circ\mathcal{D}}{\partial \boldsymbol{w}^{(l)}} \right\| \cdot \left\| e^{-t\boldsymbol{A}^{(l)}} \right\| \cdot \left\| \mathcal{E}_{\boldsymbol{\theta}}^{(l)}(\boldsymbol{x}_t), e^{(t-T)\boldsymbol{A}^{(l)}} \boldsymbol{w}_T^{(l)} \right\|_2^2, \tag{18}$$

*where $d(\cdot,\cdot)$ denotes suitable distance metric such as squared $L_2$-norm or perceptual distances like LPIPS with a corresponding feature extractor $\mathcal{F}$, i.e., $d(\boldsymbol{x}, \boldsymbol{y}) \triangleq \|\mathcal{F}(\boldsymbol{x}) - \mathcal{F}(\boldsymbol{y})\|_2^2$. The approximation holds up to small residual terms governed by the assumptions as detailed in the derivation.*

*Proof.* Our proof relies on three key assumptions. The assumptions are (1) the level-based Koopman states $\boldsymbol{w}_t^{(l)}$ are disentangled across levels labeled by $l$, (2) the mapping $\mathcal{H}$ is isotropic w.r.t. the Koopman states, and (3) the spectrums of the Koopman observables are close to 1.

Intuitively, it requires (1) Koopman states in different levels depict different features, (2) each Koopman state has the same unit and contributes equally to the loss function, and (3) the Koopman space sufficiently models the latent evolution. A detailed version of the derivation, with the divergence from the assumptions included, is

$$\mathbb{E}_{t\sim\mathcal{U}[\varepsilon,T),\boldsymbol{x}_t\sim p_{\mathrm{data}}} \left\| \mathcal{F}\circ\mathcal{D} \left[ \{e^{-t\boldsymbol{A}^{(l)}} \mathcal{E}_{\boldsymbol{\theta}}^{(l)}(\boldsymbol{x}_t)\}_{l=1}^L \right] - \mathcal{F}\circ\mathcal{D} \left[ \{\boldsymbol{w}_0^{(l)}\}_{l=1}^L \right] \right\|_2^2, \tag{19}$$

$$\overset{(*)}{=} \sum_{l=1}^L \mathbb{E} \left\| \mathcal{H}(\boldsymbol{w}_l) - \mathcal{H}(\boldsymbol{w}_{l-1}) \right\|_2^2 + o(\rho), \tag{20}$$

$$\overset{(**)}{=} \sum_{l=1}^L \mathbb{E} \left\| \frac{\partial \mathcal{H}}{\partial \boldsymbol{w}^{(l)}} \right\| \cdot \left\| e^{-t\boldsymbol{A}^{(l)}} \mathcal{E}_{\boldsymbol{\theta}}^{(l)}(\boldsymbol{x}_t) - \boldsymbol{w}_0^{(l)} \right\|_2^2 + o(\rho + \varepsilon), \tag{21}$$

$$\overset{(***)}{=} \sum_{l=1}^L \mathbb{E} \left\| \frac{\partial \mathcal{H}}{\partial \boldsymbol{w}^{(l)}} \right\| \cdot \left\| \mathcal{E}_{\boldsymbol{\theta}}^{(l)}(\boldsymbol{x}_t) - \boldsymbol{w}_t^{(l)} \right\|_2^2 + o(\rho + \varepsilon + \delta), \tag{22}$$

where parameters $\boldsymbol{w}_l \triangleq \{\mathrm{e}^{-t\boldsymbol{A}^{(l)}} \mathcal{E}_{\boldsymbol{\theta}}^{(l)}(\boldsymbol{x}_t)\}_{k=1}^{l} \cup \{\boldsymbol{w}_0^{(l)}\}_{k=l+1}^{L}$, and the equations $\overset{(*)}{=}$, $\overset{(**)}{=}$, and $\overset{(***)}{=}$ correspond to the assumptions above respectively. The error term consists of three small-value terms where $\rho$ is the extent of entanglement among levels, $\varepsilon$ is the amount of anisotropy of $\mathcal{H}$, and $\delta$ is the range of Koopman spectrum.

$\square$

**Proposition C.3** (Equivalence of Trajectory Consistency Losses under Structural Assumptions, Formal). *Under the assumptions that (1) the level-based Koopman states $\boldsymbol{w}_t^{(l)}$ are strongly disentangled across levels with an upper bound of correlations $\rho$, (2) the mapping $\mathcal{H}$ is isotropic w.r.t. the Koopman states, with a lower bound of $\eta^{(l)}$ with is non-zero weights for sufficient models, and (3) the spectrums of the Koopman observables are close to $1$, which leads to the fact that the maximal spectrum $\alpha$ of matrix $\boldsymbol{A}$ has a small positive value. Following the notations $\boldsymbol{w}^{(l)}$, $\mathcal{H}$, $\mathcal{F}$, and $d$ in Proposition C.2., we have*

$$\mathcal{L}_{t-consist} = \mathbb{E}_{t \sim \mathcal{U}[\epsilon, T]} \left[ d \left( \mathcal{D}_{\boldsymbol{\phi}}(\{e^{(\varepsilon-t)\boldsymbol{A}^{(l)}} \mathcal{E}_{\boldsymbol{\theta}}^{(l)}(\boldsymbol{x}_t)\}_{l=1}^{L}), \boldsymbol{x}_\epsilon \right) \right], \tag{23}$$

$$\geq \left( 1 - \frac{\rho^2 L}{4} \right) \cdot e^{-t\alpha} \cdot \sum_{l=1}^{L} \eta^{(l)} \cdot \mathbb{E}\|\mathcal{E}_{\boldsymbol{\theta}}^{(l)}(\boldsymbol{x}_t) - e^{(t-T)\boldsymbol{A}^{(l)}} \boldsymbol{w}_T^{(l)}\|_2^2, \tag{24}$$

*which showcases that the minimization of $\mathcal{L}_{t-consist}$ enforces the minimization of estimation error in the Koopman space.*

*Proof.* Under the given assumptions, we have

$$\mathbb{E}_{t \sim \mathcal{U}[\varepsilon, T], \boldsymbol{x}_t \sim p_{\text{data}}} \|\mathcal{F} \circ \mathcal{D}[\boldsymbol{w}_L] - \mathcal{F} \circ \mathcal{D}[\boldsymbol{w}_0]\|_2^2 = \mathbb{E} \left\| \sum_{l=1}^{L} [\mathcal{H}(\boldsymbol{w}_l) - \mathcal{H}(\boldsymbol{w}_{l-1})] \right\|_2^2, \tag{25}$$

$$= \sum_{l=1}^{L} \mathbb{E}\|\mathcal{H}(\boldsymbol{w}_l) - \mathcal{H}(\boldsymbol{w}_{l-1})\|_2^2 + \sum_{\substack{l,k=1 \\ l \neq k}}^{L} \mathbb{E}[\mathcal{H}(\boldsymbol{w}_l) - \mathcal{H}(\boldsymbol{w}_{l-1})]^{\top}[\mathcal{H}(\boldsymbol{w}_k) - \mathcal{H}(\boldsymbol{w}_{k-1})], \tag{26}$$

$$\overset{(*)}{\geq} \left( 1 - \frac{\rho^2 L}{4} \right) \sum_{l=1}^{L} \mathbb{E}\|\mathcal{H}(\boldsymbol{w}_l) - \mathcal{H}(\boldsymbol{w}_{l-1})\|_2^2, \tag{27}$$

$$\overset{(**)}{\geq} \left( 1 - \frac{\rho^2 L}{4} \right) \sum_{l=1}^{L} \eta^{(l)} \cdot \mathbb{E}\|\mathrm{e}^{-t\boldsymbol{A}^{(l)}} \mathcal{E}_{\boldsymbol{\theta}}^{(l)}(\boldsymbol{x}_t) - \boldsymbol{w}_0^{(l)}\|_2^2, \tag{28}$$

$$\overset{(***)}{\geq} \left( 1 - \frac{\rho^2 L}{4} \right) \cdot \mathrm{e}^{-t\alpha} \cdot \sum_{l=1}^{L} \eta^{(l)} \cdot \mathbb{E}\|\mathcal{E}_{\boldsymbol{\theta}}^{(l)}(\boldsymbol{x}_t) - \mathrm{e}^{(t-T)\boldsymbol{A}^{(l)}} \boldsymbol{w}_T^{(l)}\|_2^2, \tag{29}$$

where parameters $\boldsymbol{w}_l \triangleq \{\mathrm{e}^{-t\boldsymbol{A}^{(l)}} \mathcal{E}_{\boldsymbol{\theta}}^{(l)}(\boldsymbol{x}_t)\}_{k=1}^{l} \cup \{\boldsymbol{w}_0^{(l)}\}_{k=l+1}^{L}$. In addition, the inequality $\overset{(*)}{\geq}$ holds under the assumption (1) where the maximal entanglement of Koopman observables at different levels $\rho \triangleq \max_{l,k=1,\cdots,L} \left| \text{Corr}\left[\mathcal{H}(\boldsymbol{w}_l) - \mathcal{H}(\boldsymbol{w}_{l-1}), \mathcal{H}(\boldsymbol{w}_k) - \mathcal{H}(\boldsymbol{w}_{k-1})\right] \right|$ is small. On the other hand, the inequality $\overset{(**)}{\geq}$ and $\overset{(***)}{\geq}$ holds when $\eta^{(l)} \triangleq \inf_{\Delta\boldsymbol{w}} \|\mathcal{H}(\boldsymbol{w}^{(l)} + \Delta\boldsymbol{w}) - \mathcal{H}(\boldsymbol{w}^{(l)})\|_2^2 / \|\Delta\boldsymbol{w}\|_2^2$ is the minimal spectrum of $\mathcal{H}$, and $\alpha \triangleq r(\boldsymbol{A}^{(l)})$ is the spectral radius of matrix $\boldsymbol{A}^{(l)}$.

Under the assumptions, there holds $\rho \to 0^+$ and $\alpha \to 0^+$. Consequently, minimizing the left-hand side (as in the proposed algorithm) is equivalent to the minimization of Koopman observables to the right-hand side. $\square$

## C.3 Formal Statement and Proof of Theorem 3.1

In this section, we provide the formal statement and proof of Theorem 3.1. Before that, we first give Definition C.1, Proposition C.3, and Lemma C.1 to introduce Theorem 3.1.

In order to illustrate the feasibility of our proposed method, we propose to measure the complexity of functions by the estimation accuracy using the simplicial complex created by Delaunay triangulation.

Def. C.1 first introduces the simplicial error, which reflects the complexity of the subset $\Omega$ as $\varepsilon_N(\Omega) = 0$ if and only if $\dim\Omega = m$ and $\Omega$ is convex.

**Definition C.1** (The Simplicial Error). *For a compact subset of an $m$-dimensional manifold $\Omega \subset \mathcal{M}^m \subset \mathbb{R}^d$, the simplicial error of it is $\varepsilon_N(\Omega) \triangleq \inf_{\Xi \subset \Omega,\ |S_m(\Xi)|=N} \varepsilon_\Xi(\Omega)$ where $\Xi$ is a set of $N$ generic point samples in $\Omega$, $\varepsilon_\Xi(\Omega) \triangleq \sup_{x \in \Omega} \min_{s \in S_m(\Xi)} d(x,s)$ is the error of the $m$-dimensional Delaunay triangulation $S_m(\Xi)$ of the point set $\Xi$, and $d(x,s) \triangleq \inf_{y \in s}\|x-y\|$ is the minimal distance from $x$ to simplex $s$. The norm $\|\cdot\|$ is commonly $L_2$-norm.*

We then propose to measure the theoretical error bounds for network estimation by the error between its graph and a closest simplicial complex. Firstly, Prop. C.4 defines the function complexity, which is intuitively influenced by the complexity of its domain and range sets. The conclusion in Lem. C.1 further shows a bound for the estimation error of the function by a network using ReLU activations. The error is closely related to the function complexity we defined, which is coherent with the common knowledge, as the network creates a simplicial complex with each activation creating a surface between simplicials.

**Proposition C.4.** *The function simplicial error is $\varepsilon_N(f) \triangleq \varepsilon_N(\{(\boldsymbol{x}, f(\boldsymbol{x}))|\boldsymbol{x} \in \Omega\})$ for $f : \Omega \to \Gamma$. We then have $\max\{\varepsilon_N(\Omega), \varepsilon_N(\Gamma)\} \le \varepsilon_N(f) \le \inf_{\Xi \subset \Omega,\ |S_m(\Xi)|=N}\|[\varepsilon_\Xi(\Omega), \varepsilon_\Xi(\Gamma)]\|$.*

*Proof.* We first prove the left side.

$$\varepsilon_N(f) = \inf_{\substack{\Xi \subseteq \Theta \\ |S_m(\Xi)|=N}} \left[\sup_{\theta \in \Theta} \min_{s \in S_m(\Xi)} d(\theta, s)\right] \ge \inf_{\substack{\Xi \subseteq \Theta \\ |S_m(\Xi)|=N}} \left[\sup_{\theta \in \Theta} \min_{s \in S_m(\Xi)} d(P_\Omega\theta, P_\Omega s)\right],$$

$$= \inf_{\substack{P_\Omega\Xi \subset P_\Omega\Theta \\ |S_m(P_\Omega\Xi)|=N}} \left[\sup_{P_\Omega\theta \in P_\Omega\Theta} \min_{P_\Omega s \in S_m(P_\Omega\Xi)} d(P_\Omega\theta, P_\Omega s)\right] = \varepsilon_N(P_\Omega\Theta) = \varepsilon_N(\Omega),$$

where $P_\Omega$ is the projector to $\Omega$. Similarly,

$$\varepsilon_N(f) = \inf_{\substack{\Xi \subseteq \Theta \\ |S_m(\Xi)|=N}} \left[\sup_{\theta \in \Theta} \min_{s \in S_m(\Xi)} d(\theta, s)\right] \ge \inf_{\substack{\Xi \subseteq \Theta \\ |S_m(\Xi)|=N}} \left[\sup_{\theta \in \Theta} \min_{s \in S_m(\Xi)} d(P_\Gamma\theta, P_\Gamma s)\right],$$

$$= \inf_{\substack{P_\Omega\Xi \subset \Omega \\ |S_m(P_\Omega\Xi)|=N}} \left[\sup_{P_\Gamma\theta \in P_\Gamma\Theta} \min_{P_\Gamma s \in S_m[f(P_\Omega\Xi)]} d[P_\Gamma\theta, P_\Gamma s]\right],$$

$$= \inf_{\substack{\tilde{\Xi} \subseteq \Omega \\ |S_m(\tilde{\Xi})|=N}} \varepsilon_{f(\tilde{\Xi})}(\Gamma) \ge \varepsilon_N(\Gamma).$$

On the other hand, the right-hand side holds due to,

$$\varepsilon_N(f) = \inf_{\substack{\Xi \subseteq \Theta \\ |S_m(\Xi)|=N}} \left[\sup_{\theta \in \Theta} \min_{s \in S_m(\Xi)} \inf_{\varphi \in s} \left\|[\|P_\Omega(\theta - \varphi)\|, \|P_\Gamma(\theta - \varphi)\|]\right\|\right],$$

$$\le \inf_{\substack{\Xi \subseteq \Theta \\ |S_m(\Xi)|=N}} \left[\sup_{\theta \in \Theta} \min_{s \in S_m(\Xi)} \inf_{\varphi \in s} \left\|[\varepsilon_\Xi(P_\Omega\Theta), \varepsilon_\Xi(P_\Gamma\Theta)]\right\|\right],$$

$$\overset{\star}{=} \inf_{\substack{\tilde{\Xi} \subseteq \Omega \\ |S_m(\tilde{\Xi})|=N}} \left\|[\varepsilon_{\tilde{\Xi}}(\Omega), \varepsilon_{f(\tilde{\Xi})}(\Gamma)]\right\|,$$

where $\overset{\star}{=}$ holds as the norm inside $\inf$ is a constant w.r.t. $\theta, \varphi$ and $s$. $\square$

**Lemma C.1.** *The estimation of function $f : \Omega^m \to \Gamma$ by a network $\mathcal{F}$ with $n_a$ ReLU activations satisfies $\mathbb{P}\left(\|\mathcal{F}(x) - f(x)\| \le \varepsilon_{\lceil 2n_a/(m+1)\rceil}(f)\right) \ge 1 - \delta$ where $\delta$ indicates the amount undertrained.*

*Proof.* With reference to the universal approximation theorem, each $(m-1)$- dimensional face of the simplex matches a cuspidal point of a ReLU activation. For a network with $n_a$ activations, the maximal number of faces is $n_a$ which means there are at most $\lceil \frac{2n_a}{m+1}\rceil$ simplices in the simplicial complex, as each simplex has $m+1$ faces and all faces are clamped between (exactly) two simplices.

For activations other than ReLU, one may use intrinsic Delaunay triangulation to replace the Delaunay triangulation. $\square$

**Theorem C.1** (Formal Statement of Theorem 3.1). *If the noise space $\boldsymbol{x}_T \in \Xi \subset \mathbb{R}^{n\times n}$ and the Koopman space $\Psi \subset \mathbb{R}^D$ are compact, the ideal estimation error of our proposed HKD model with encoder $\mathcal{E}$ and decoder $\mathcal{D}$ networks of $n_a/2$ activation functions is smaller than that of an end-to-end one-step model $\mathcal{F}$ with $n_a$ activation functions, i.e., $err_{HKD} \leq err_{one\text{-}step} + O(\kappa)$ holds for any one-step diffusion where*

$$err_{HKD} = \varepsilon_{\lceil n_a/(n^2+1)\rceil}(\mathcal{E}) + \kappa + \varepsilon_{\lceil n_a/(D+1)\rceil}(\mathcal{D}) \; and \; err_{one\text{-}step} = \varepsilon_{\lceil 2n_a/(n^2+1)\rceil}(\mathcal{F})$$

*for the one-step mapping $\mathcal{F}$, the encoder $\mathcal{E}$ and decoder $\mathcal{D}$ in the proposed framework. The additional Koopman error $\kappa$ is*

$$\kappa \triangleq \max\left\{ \frac{2\sqrt{3}\sigma m^2 \rho_{\sup}}{\sqrt{N(\frac{\delta}{3}-2m^{-\frac{r}{3}})}-\sqrt{3}\sigma m^2}, \frac{2\sqrt{3}\sigma m^2 \rho_{\inf}}{\sqrt{N(\frac{\delta}{3}-2m^{-\frac{r}{3}})\cdot\rho_{\inf}}-\sqrt{3}\sigma m^2} \right\} + o(m^{-\frac{r}{3}}),$$

*which is small when the Koopman dimension $m$ is sufficient and the dataset size $N$ is large enough. Here, parameter $\sigma$ is the standard deviation of the observables, $\delta$ is $0$ when the network is perfectly trained, $(\rho_{\inf}, \rho_{\sup})$ is the true spectral range of the Koopman operator and $r \to -\infty$ when the $m$-dimensional Koopman process adequately model the Koopman space.*

*Proof.* Under the assumption of AE-based data compression methods, we assume that the data are approximately on an $m$-dimensional manifold $\mathcal{M}^m \subset \mathbb{R}^{n\times n}$ defined by a bijection $\mathfrak{F}: \mathcal{M} \leftrightarrow \mathbb{R}^m$, and that the projection of the dataset on the manifold is $\Omega \subset \mathcal{M}$. Let $\Psi = \mathfrak{F}(\Omega)$ be a compact subset of $\mathbb{R}^m$ that encodes the dataset.

We consider $\boldsymbol{x}_t(p)$ as a function w.r.t. spatial position $p$ and define operators $\mathcal{F}_t : \boldsymbol{x}_t(\cdot) \mapsto f(\boldsymbol{x}_t(\cdot), t)$, $\mathcal{G}_t : \boldsymbol{x}_t(\cdot) \mapsto g^2(t) \cdot \nabla \log p_t(\boldsymbol{x}_t(\cdot))$, hence we have,

$$\delta(\mathfrak{F}\boldsymbol{x}_t) = [\mathfrak{F}\mathcal{F}_t\boldsymbol{x}_t - \mathfrak{F}\mathcal{G}_t\boldsymbol{x}_t]\,\delta t = \mathfrak{F}(\mathcal{F}_t - \mathcal{G}_t)\mathfrak{F}^{-1}(\mathfrak{F}\boldsymbol{x}_t)\delta t,$$

which satisfies the definition of a Koopman process where the operator is $\mathcal{K}_t = \mathcal{I} + \mathfrak{F}(\mathcal{F}_t - \mathcal{G}_t)\mathfrak{F}^{-1}$.

Letting $\boldsymbol{\xi} = (\mathcal{K}_t - \hat{\boldsymbol{K}}_t)\boldsymbol{y}_t$ for the observables. We have the bound that

$$\mathbb{P}\left( \|(\mathcal{K}_t - \hat{\boldsymbol{K}}_t)\boldsymbol{y}_t\| \leq \kappa_t \|\boldsymbol{y}_t\| \right) \geq 1 - \delta,$$

where $\kappa_t = 2\sqrt{3}\,\sigma m^2 r(\mathcal{K}_t) \big/ \left[ \sqrt{N(\delta - 2m^{-\frac{r}{3}})} \cdot \min\{1, r(\mathcal{K}_t)\} - \sqrt{3}\,\sigma m^2 \right] + o(m^{-\frac{r}{3}})$.

Here, the bound factor $\kappa_t$ is related to time $t$ while it has a uniform bound factor,

$$\kappa^{(\delta)} \triangleq \sup_{t\in[0,T]} \kappa_t = \max\left\{ \frac{2\sqrt{3}\sigma m^2 \rho_{\sup}}{\sqrt{N(\delta-2m^{-\frac{r}{3}})}-\sqrt{3}\sigma m^2}, \frac{2\sqrt{3}\sigma m^2 \rho_{\inf}}{\sqrt{N(\delta-2m^{-\frac{r}{3}})\cdot\rho_{\inf}}-\sqrt{3}\sigma m^2} \right\} + o(m^{-\frac{r}{3}}),$$

where $\rho_{\inf} \triangleq \inf_t r(\mathcal{K}_t)$ and $\rho_{\sup} \triangleq \sup_t r(\mathcal{K}_t)$.

The estimated Koopman model with an encoder $\mathcal{E}$ estimating mapping $\mathfrak{F}$ and a decoder $\mathcal{D}$ estimating $\mathfrak{F}^{-1}$ (with each of them having $n_a/2$ activations) has a probabilistic error bound,

$$\mathbb{P}\left( \|\mathcal{D}\boldsymbol{z}_0 - \boldsymbol{x}_0\| \leq \varepsilon \right), \text{ where } \boldsymbol{z}_t = \boldsymbol{z}_T + \int_T^t (\hat{\boldsymbol{K}}_s - \boldsymbol{I})\boldsymbol{z}_s\,\mathrm{d}s, \; \boldsymbol{z}_T = \mathcal{E}(\boldsymbol{x}_T),$$

$$\text{and } \boldsymbol{x}_0 = \mathfrak{F}^{-1}\boldsymbol{y}_0, \; \boldsymbol{y}_t = \boldsymbol{y}_T + \int_T^t (\mathcal{K}_s - \mathcal{I})\boldsymbol{y}_s\,\mathrm{d}s, \; \boldsymbol{y}_T = \mathfrak{F}\boldsymbol{x}_T,$$

$$\geq \mathbb{P}(\|\mathcal{D}\boldsymbol{z}_0 - \mathfrak{F}^{-1}\boldsymbol{z}_0\| + \|\mathfrak{F}^{-1}\boldsymbol{z}_0 - \mathfrak{F}^{-1}\boldsymbol{y}_0\| \leq \varepsilon)$$

$$\overset{\star}{\geq} \mathbb{P}(\|\mathcal{D}\boldsymbol{z}_0 - \mathfrak{F}^{-1}\boldsymbol{z}_0\| \leq \varepsilon_{\lceil\frac{n_a}{D+1}\rceil}(\mathfrak{F}^{-1})) \cdot \mathbb{P}(\|\boldsymbol{z}_0 - \boldsymbol{y}_0\| \leq \kappa^{(\delta/3)}\zeta(\mathrm{e}^{\rho T} - 1)/\rho)$$

$$\cdot \mathbb{P}(\|\boldsymbol{z}_T - \boldsymbol{y}_T\| \leq \varepsilon_{\lceil\frac{n_a}{n^2+1}\rceil}(\mathfrak{F})),$$

$$\geq (1 - \delta/3)^3 \geq 1 - \delta.$$

Here, $\overset{*}{\geq}$ holds due to Lemma 1 and $\overset{\star}{\geq}$ holds when

$$\varepsilon = \varepsilon_{\lceil\frac{n_a}{D+1}\rceil}(\mathfrak{F}^{-1}) + r(\mathfrak{F}^{-1}) \cdot \kappa^{(\delta/3)}\zeta(\mathrm{e}^{\rho T} - 1)/\rho + r(\mathfrak{F}^{-1}) \cdot \mathrm{e}^{\rho T} \cdot \varepsilon_{\lceil\frac{n_a}{n^2+1}\rceil}(\mathfrak{F}),$$

where $r(\mathfrak{F}^{-1})$ is the spectrum radius of $\mathfrak{F}^{-1}$ and the bound for $\|\boldsymbol{z}_0 - \boldsymbol{y}_0\|$ is given by

$$\|\boldsymbol{z}_t - \boldsymbol{y}_t\| \leq \|\boldsymbol{z}_T - \boldsymbol{y}_T\| + \left\|\int_T^t (\mathcal{K}_s - \mathcal{I})(\boldsymbol{z}_s - \boldsymbol{y}_s)\mathrm{d}s\right\| + \int_T^t \|(\hat{\boldsymbol{K}}_s - \mathcal{K}_s)\boldsymbol{z}_s\|\mathrm{d}s,$$

$$\leq \|\boldsymbol{z}_T - \boldsymbol{y}_T\| + \rho \int_t^T \|\boldsymbol{z}_s - \boldsymbol{y}_s\|\mathrm{d}s + \kappa\zeta(T - t),$$

where $\rho \triangleq \max\{1 - \rho_{\mathrm{inf}}, \rho_{\mathrm{sup}} - 1\}$ and $\zeta$ is a bound of $\|\boldsymbol{z}_t\|$. Let $D_t$ satisfies $D_T = \|\boldsymbol{z}_T - \boldsymbol{y}_T\|$ and $\frac{\mathrm{d}D_t}{\mathrm{d}t} = -\rho D_t - \kappa\zeta$ which leads to $d_t \triangleq \|\boldsymbol{z}_t - \tilde{\boldsymbol{z}}_t\| \leq D_t$ as they are positive trajectories. The solution to $D_t$ is $D_t = \frac{\kappa\zeta}{\rho}\left[\mathrm{e}^{\rho(T-t)} - 1\right] + D_T\mathrm{e}^{\rho(T-t)}$.

For ideal $\mathcal{K}_t$ and $\mathfrak{F}$, $\varepsilon$ is dominated by $\varepsilon_{\lceil \frac{n_a}{D+1}\rceil}(\mathfrak{F}^{-1}) + \varepsilon_{\lceil \frac{n_a}{n^2+1}\rceil}(\mathfrak{F})$ as $\rho$ is small and $r(\mathfrak{F}^{-1}) \approx 1$.

On the other hand, the one-step methods aim at estimating $\hat{\boldsymbol{x}}_0 = \mathcal{F}(\boldsymbol{x}_T) \triangleq \boldsymbol{x}_T + \int_T^0 (\mathcal{F}_t - \mathcal{G}_t)\boldsymbol{x}_t\mathrm{d}t$ by a network $\mathcal{F}$ with $n_a$ activation functions. The error satisfies, according to Lemma 1,

$$\mathbb{P}\left(\|\mathcal{F}(\boldsymbol{x}_T) - \boldsymbol{x}_0\| \leq \varepsilon_{\lceil 2n_a/(n^2+1)\rceil}(\mathcal{F})\right) \geq 1 - \delta.$$

Note that ideally, $\boldsymbol{x}_T \in \Xi \subset \mathbb{R}^{n\times n}$, $\boldsymbol{x}_0 \in \Omega \subset \mathcal{M}^m$ and $\mathfrak{F}\boldsymbol{x}_0 \in \Psi \subset \mathbb{R}^m$, where $\Xi$ and $\Lambda$ are compact, hence

$$\varepsilon_{\lceil \frac{2n_a}{n^2+1}\rceil}(\mathcal{F}), \tag{30}$$

$$\geq \max\left\{\varepsilon_{\lceil \frac{2n_a}{n^2+1}\rceil}(\Xi), \varepsilon_{\lceil \frac{2n_a}{n^2+1}\rceil}(\Omega)\right\}, \qquad \Leftarrow \text{Proposition 1} \tag{31}$$

$$= \varepsilon_{\lceil \frac{2n_a}{n^2+1}\rceil}(\Omega), \qquad \Leftarrow \Omega \text{ is compact} \tag{32}$$

$$= \|[\varepsilon_{\mathfrak{F}(\Delta)}(\Psi), \varepsilon_\Delta(\Omega)]\| + \|[\varepsilon_{\cdot}(\Xi), \varepsilon_{\mathfrak{F}(\cdot)}(\Psi)]\|, \qquad \Leftarrow \Xi \text{ and } \Psi \text{ are compact} \tag{33}$$

$$\text{where } \Delta \text{ satisfies } \varepsilon_\Delta(\Omega) = \varepsilon_{\lceil \frac{2n_a}{n^2+1}\rceil}(\Omega), \qquad \varepsilon_{\cdot}(\Xi) = \varepsilon_{\cdot}(\Psi) = 0 \tag{34}$$

$$\geq \varepsilon_{\lceil \frac{n_a}{D+1}\rceil}(\mathfrak{F}^{-1}) + \varepsilon_{\lceil \frac{n_a}{n^2+1}\rceil}(\mathfrak{F}). \qquad \Leftarrow \text{Proposition 1} \tag{35}$$

$$\tag{36}$$

Using Lem. C.1, we may reach the conclusion in Thm. C.1, which shows that if the neural network models are trained properly, the theoretical error bound of the proposed method is smaller than that of any other end-to-end one-step method. $\qquad\square$

## D  Experimental Supplement

### D.1  Additional Experimental Results for Sec. 4.1

**Conditional Generation on CIFAR-10.**  In this part, we provide the conditional generation results on CIFAR-10 to show the generalization ability of our framework on class-conditional tasks. The results are provided in Tab. 4.

Table 4: Class-conditional sample quality on CIFAR-10 dataset.

| Methods | NFE($\downarrow$) | FID($\downarrow$) |
|---|---|---|
| Score SDE | 2000 | 2.20 |
| EDM | 35 | 1.79 |
| DMD (w/o REG) | 1 | 5.58 |
| DMD (w/o KL) | 1 | 3.82 |
| DMD | 1 | 2.66 |
| HKD | 1 | 2.77 |

**Visual Results.**  In this section, we provide additional qualitative results on CIFAR-10 (unconditional), FFHQ (unconditional), and CIFAR-10 (conditional) to further demonstrate the visual quality and effectiveness of our proposed method. The results are presented in Fig. 7, Fig. 8, Fig. 9, respectively.

**Training Stability Results.**  We include the results of training HKD five times independently on the CIFAR-10 dataset in Tab. 5. The FID scores and their standard deviation are reported in the table below. We summarize the FID results over every 10 epochs during training across the five runs, demonstrating stable convergence behavior.

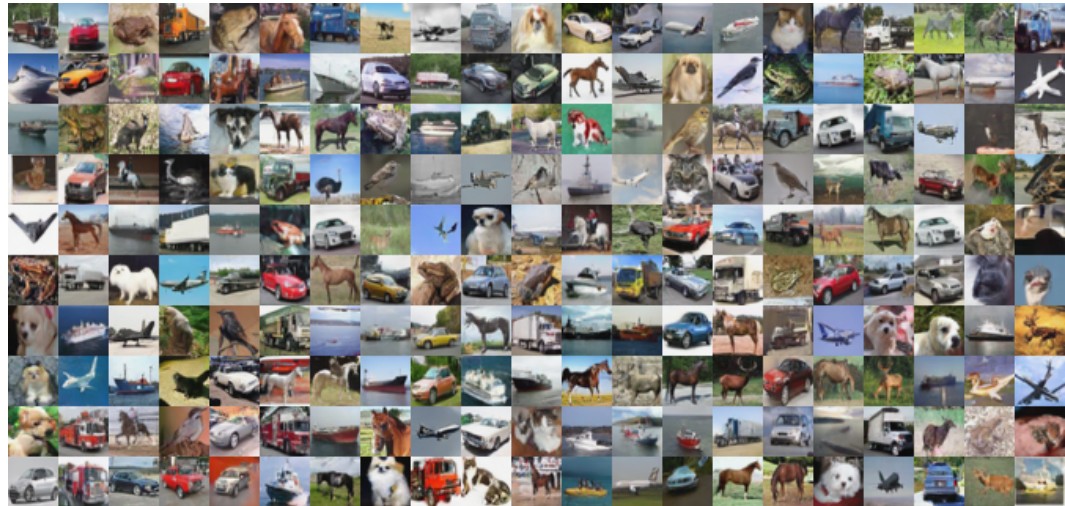

Figure 7: The additional visualization of image generations from HKD trained by the CIFAR10 dataset.

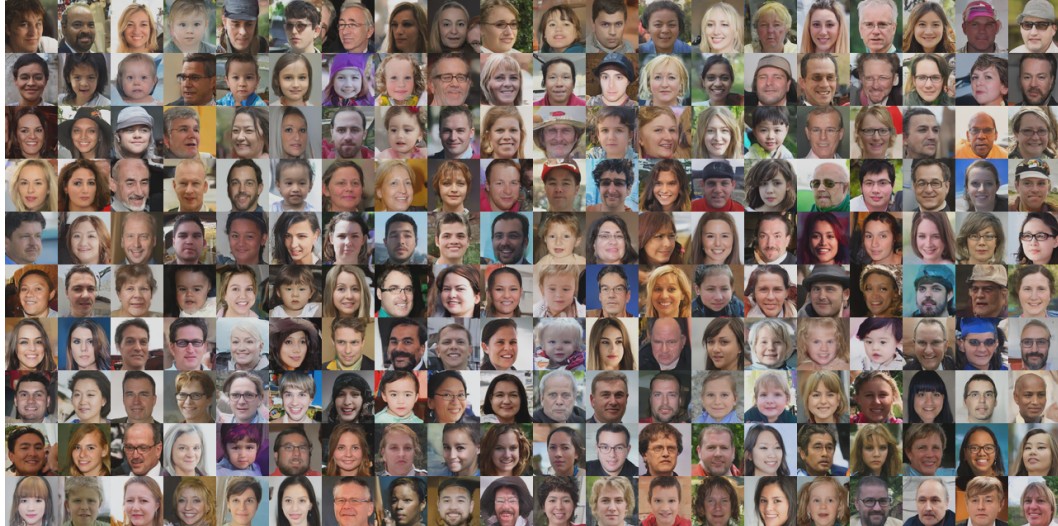

Figure 8: The additional visualization of image generations from HKD trained by the FFHQ dataset.

**The Wall-clock Time Per Image for Generation Methods.** We provide the wall-clock time per image for different generation methods in Tab. 6, measured on an NVIDIA V100 GPU. These methods were trained on the CIFAR-10 dataset.

### D.2 Additional Experimental Results for Sec. 4.2

We provide additional visualization results of image models on the CIFAR-10 dataset in Fig. 10, and on the FFHQ dataset in Fig. 11. As shown in Fig. 11, the reconstructed images using low-frequency components primarily capture the semantic structure of the face. The mid-frequency components contribute to the overall contour and shape details of the face, while the high-frequency components are responsible for fine-grained details such as facial hair.

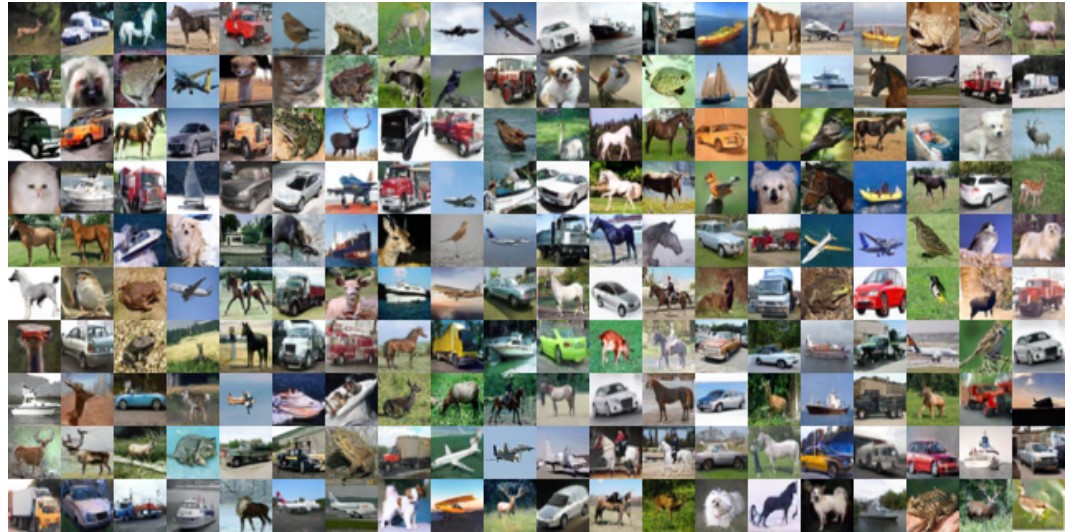

Figure 9: The additional visualization of conditional image generations from HKD trained by the CIFAR10 dataset.

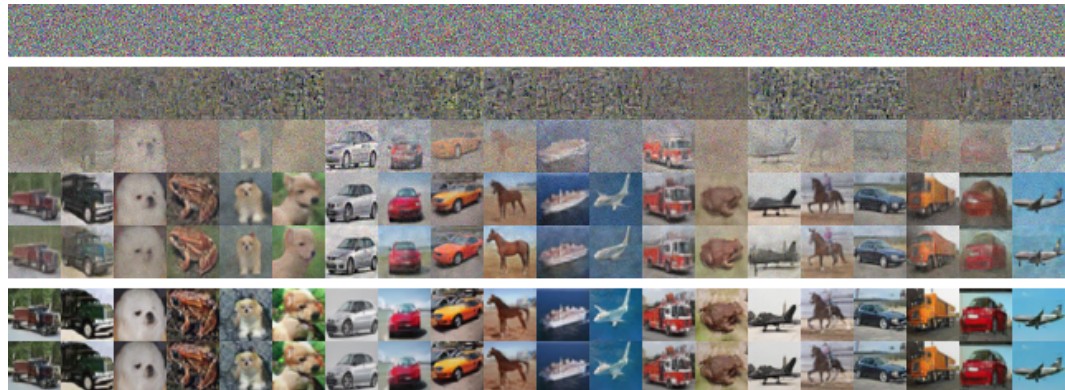

Figure 10: The additional visualization of image modes for the CIFAR10 dataset. The rows correspond to those in the manuscript.

## D.3 Additional Experimental Results for Sec. 4.3

**Additional Results on One-step Image Editing via Frequency-aware Interventions.** We provide the additional results on the one-step image editing experiment via frequency-aware interventions along the diffusion trajectory. The results are provided in Fig. 12, 13.

## D.4 More Implementation Details about Experiments for Sec. 4.1

**Network Architecture.** In our experiment, both the encoder and decoder are designed to follow the architecture of the EDM [8] encoder and decoder, leveraging the proven effectiveness of existing diffusion model designs, similar to consistency models [32]. Inspired by ECM [4], we utilize EDM's pre-trained weights for our encoder and decoder modules to provide structured latent-to-output mappings and prevent mode collapse, retaining rich hierarchical features acquired during large-scale training. For each linear operator $\boldsymbol{A}^{(l)}(i, j)$, which adopts a block-diagonal structure, we perform vectorization by concatenating the real and imaginary parts—placing the real components in the first half of the vector and the imaginary components in the second half. The optimization is then conducted over the elements of this vectorized form. We train our model end-to-end.

Table 5: FID scores (mean ± standard deviation) during training across epochs.

|  | Epoch 0 | Epoch 10 | Epoch 20 |
|---|---|---|---|
| **FID (Mean ± Std)** | 11.82±0.29 | 5.70±0.21 | 4.71±0.17 |
|  | **Epoch 30** | **Epoch 40** | **Epoch 50** |
| **FID (Mean ± Std)** | 4.09±0.18 | 3.52±0.11 | 3.31±0.03 |
|  | **Epoch 52** | **Epoch 53** | **Before Training** |
| **FID (Mean ± Std)** | 3.31±0.02 | 3.30±0.02 | 447.13 |

Table 6: NFE and per-image latency across different generation methods.

|  | DDPM | Score SDE | DDIM | DDIM |
|---|---|---|---|---|
| **NFE** | 1000 | 2000 | 10 | 100 |
| **Latency** | 15.32s | 45.89s | 0.51s | 2.35s |
|  | **EDM** | **EDM** | **KD** | **PD** |
| **NFE** | 35 | 15 | 1 | 1 |
| **Latency** | 0.69s | 0.30s | 0.26s | 0.28s |
|  | **CD (LPIPS)** | **DMD** | **RF-1** | **RF-2** |
| **NFE** | 1 | 1 | 1 | 1 |
| **Latency** | 0.02s | 0.03s | 0.03s | 0.03s |
|  | **RF-3** | **RF-2++** | **CT (LPIPS)** | **iCT** |
| **NFE** | 1 | 1 | 1 | 1 |
| **Latency** | 0.03s | 0.03s | 0.02s | 0.02s |
|  | **iCT-deep** | **ECM** | **HKD** | — |
| **NFE** | 1 | 1 | 1 | — |
| **Latency** | 0.02s | 0.02s | 0.03s | — |

Compared to conventional diffusion models, our framework introduces only one additional parameter component, the linear operator $A$. However, due to the structured block-diagonal assumption imposed on $A$, the overall increase in parameter count is minimal.

In addition, the EDM U-Net architecture supports augment_label as an input, which we set to None. Additionally, as in EDM, the U-Net receives the noise scale $\sigma$ as an input. Although the observable functions in our model are inherently time-independent, we follow the EDM convention and feed the time-dependent $\sigma(t)$ into the encoder to maintain architectural compatibility and enhance representational capacity. In contrast, our decoder only operates on the final timestep; therefore, it consistently receives $\sigma(t = 1)$ as input. The EDM architecture also allows for class_labels as input. For unconditional generation tasks on CIFAR-10 and FFHQ, we set class_labels = None. For conditional generation on CIFAR-10, we adopt the same label configuration as in EDM.

**Hyperparameter Setting.** We conduct experiments using 8 NVIDIA V100 GPUs, with a batch size of 256 for the CIFAR-10 dataset and 64 for the FFHQ dataset. The loss function weights are set as $\lambda_2 = 1$ and $\lambda_1 = 10^{-3^{(\text{current\_epoch}/\text{overall\_epoch})}}$, where $\lambda_1$ decays exponentially over the course of training. For the implementation of the trajectory consistency loss, we employ a Monte Carlo sampling strategy: in each training iteration, four intermediate timesteps are randomly sampled uniformly, and the loss is computed as the average over these sampled time points.

### D.5 More Details about Comparison Methods

To provide a comprehensive comparison, we categorize the baseline methods into three groups: multi-step diffusion models, distillation-based one-step models, and consistency-based models. Below, we summarize the key characteristics and mechanisms of each method.

**Multi-step Diffusion Baselines.** 1. DDPM (Denoising Diffusion Probabilistic Models) [7] A foundational diffusion model that learns to reverse a fixed Markovian noising process via iterative

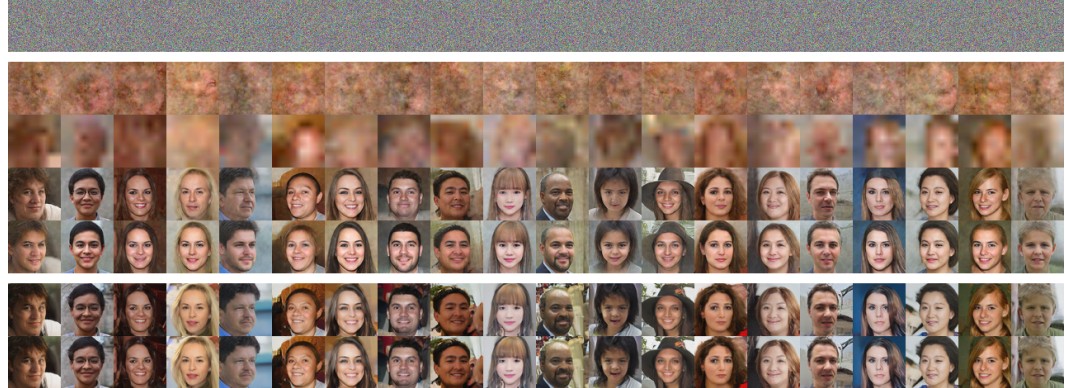

Figure 11: The visualization of image modes for the FFHQ dataset. The rows correspond to those in the manuscript. The third row, indicating the low-frequency semantics, underwent a downsampling to avoid the uncanny valley effect.

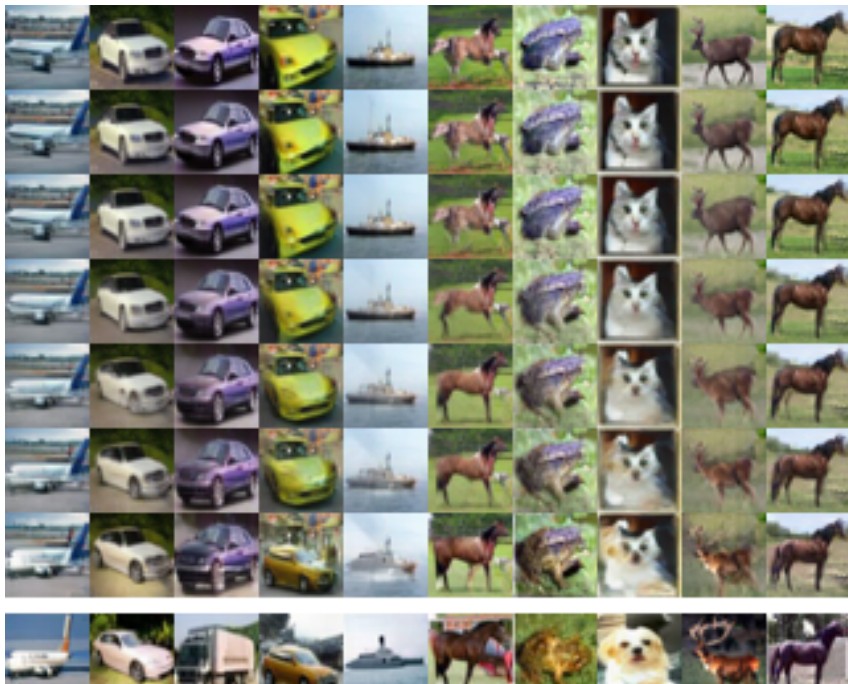

Figure 12: The visualization of frequency-aware interventions of images from CIFAR10. The rows correspond to the columns in the manuscript, meaning the original image for row 1, frequency-aware editing at factors of 10%, 20%, 50%, 80%, and 90%, frequency-agnostic editing at a factor of 90%, and the reference image, respectively.

denoising. The model is trained using a mean squared error (MSE) loss between predicted and actual noise. Sampling typically requires hundreds to thousands of steps.

2. Score SDE (Score-based Generative Modeling via Stochastic Differential Equations) [33] A Generalization towards continuous time via stochastic differential equations (SDEs). The model is trained via score matching and enables sampling using reverse-time SDE or ODE solvers, improving sample quality and flexibility.

3. DDIM (Denoising Diffusion Implicit Models) [30] A non-Markovian and deterministic variant of DDPM that introduces a sampling ODE, enabling fast sampling with fewer steps while preserving high visual fidelity. It is compatible with pre-trained DDPM models without requiring retraining.

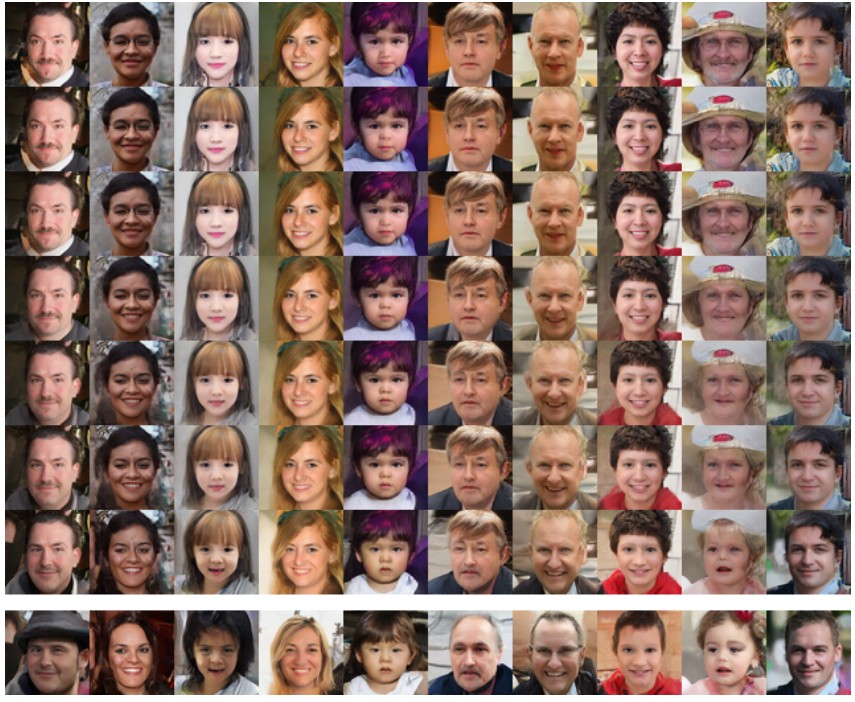

Figure 13: The visualization of frequency-aware interventions of images from FFHQ. The rows correspond to the columns in the manuscript.

4. EDM (Elucidating the Design Space of Diffusion Models) [8] A state-of-the-art diffusion framework with improved noise scheduling (log-normal distribution) and score network preconditioning. It achieves superior performance across datasets and serves as the backbone in our proposed framework.

**Distillation-based One-step Models.** 1. KD (Knowledge Distillation) [17] A straightforward distillation approach where a student model learns to replicate the output of a pre-trained diffusion teacher via MSE loss. While efficient, performance can degrade with large step-size reductions.

2. PD (Progressive Distillation) [27] Gradually distills a high-step teacher into a low-step student via intermediate models, reducing the number of steps incrementally. This approach improves stability and fidelity over direct distillation.

3. CD (LPIPS) (Consistency Distillation with Perceptual Loss) [32] Extends distillation by employing perceptual similarity metrics (LPIPS) as the training objective. It improves visual quality by preserving perceptual features across time steps.

4. DMD (Distribution Matching Distillation) [39] Aims to directly match the student's output distribution to that of the teacher using divergence-based objectives (e.g., KL divergence). This results in more accurate distribution alignment and improved sample diversity.

5. ReFlow Series (Rectified Flow and Distilled Variants) [14, 13] Utilizes a rectified flow framework to optimize ODE trajectories for efficient sampling. Variants such as 1-, 2-, and 3-Rectified Flow, and 2-Rectified Flow++ offer different trade-offs between speed and quality. Distilled versions further reduce sampling cost to 1 step with minimal performance loss.

**Consistency-based Models.** 1. CT / iCT / iCT-deep (Consistency Training) [32, 31] CT models are trained to produce consistent outputs when given different noise levels. Improved variants (iCT and iCT-deep) enforce deeper consistency across time steps, using perceptual and cosine similarity losses to enhance training stability and sample quality.

2. ECM (Easy Consistency Models) [4] Simplifies consistency training by initializing model parameters with those from a pretrained score-based model. This reduces training complexity and improves convergence without requiring complex schedule tuning.

# E Broader Impact

The proposed Hierarchical Koopman Diffusion (HKD) framework introduces a novel approach to balancing sampling efficiency and interpretability in generative modeling, which may have a significant and multifaceted societal impact. By enabling one-step image synthesis with transparent generative dynamics, HKD has the potential to make image generation systems more accessible, controllable, and explainable — attributes that are essential for responsible deployment in sensitive domains such as design, and scientific simulation.

The explicit modeling of generative trajectories and access to intermediate states could enhance human-AI collaboration, enabling users to guide or edit synthetic content with semantic intent, thus reducing risks of undesired generation and improving user trust. Furthermore, the spectral interpretability introduced by our framework can offer insights into the internal mechanics of generative models, fostering research in model debugging, safety validation, and fairness auditing.

In the broader landscape, our work contributes to the growing field of explainable generative AI, reinforcing the possibility of building models that are not only performant but also transparent and controllable. We hope this inspires further research toward interpretable, reliable, and socially responsible generative systems.

