# OpenReview forum: "Hierarchical Koopman Diffusion: Fast Generation with Interpretable Diffusion Trajectory"
_NeurIPS.cc/2025/Conference — NeurIPS 2025 poster_

### Official Review · Reviewer_xGGM · 2025-06-20

**Clarity:** 3
**Significance:** 4
**Originality:** 4
**Rating:** 6
**Confidence:** 4

**Summary:**

This paper try to remove the iterative process in the diffusion models by utilizing koopman dynamics in the hierarchical latent space of the diffusion models. In specific, they introduced a encoder that encode the latent from UNet to koopman subspace and then utilize the Koopman dynamic model to skip the intermediate timesteps to get the latent for the decoder. This enabled lower latency image editing. The experiment results shows close to best performance while the proposed method does not need to tune the hyperparameters.

**Questions:**

1. It is beneficial for the audience that the authors include the latency analysis (i.e. a table with different methods and wall clock time for each image to generate/edited)
2. Also, I suggest the authors to include experiments on PIE-Bench[1], which is a benchmark dataset for evaluating image editing performance. This helps the audience to understand the effectiveness of the proposed method for image editing task.
3. I would like the author to further clarify the proof laid out in appendix C.2, especially from line 14 to line 15 where what kind of assumption you use to establish the equivalence. Also line 16 and 18, how to understand the equivalence between these two. This question is pivotal for me since I am willing to change my score to strong accept if I the authors fully address my concerns and the proof is verified.

**Ethical Concerns:**

["NO or VERY MINOR ethics concerns only"]

**Final Justification:**

The authors has addressed my concerns and I like this paper. The paper is technically sound and has the potential to have broader impact to the community.

**Limitations:**

Yes.

**Quality:**

4

**Strengths And Weaknesses:**

Strength:

1. The paper shows an alternative way of doing one-step generation for diffusion model. The idea is novel and could lead to broader impact.
2. The paper is easy to follow and clear.

Weakness:
1. It is beneficial for the audience that the authors include the latency analysis (i.e. a table with different methods and wall clock time for each image to generate/edited)
2. Also, I suggest the authors to include experiments on PIE-Bench[1], which is a benchmark dataset for evaluating image editing performance. This helps the audience to understand the effectiveness of the proposed method for image editing task.
3. I would like the author to further clarify the proof laid out in appendix C.2, especially from line 14 to line 15 where what kind of assumption you use to establish the equivalence. Also line 16 and 18, how to understand the equivalence between these two.


[1]: Ju, Xuan, et al. "Direct inversion: Boosting diffusion-based editing with 3 lines of code." arXiv preprint arXiv:2310.01506 (2023).

---

> ### Author Rebuttal · Authors · 2025-07-31
>
> Thank you sincerely for your thoughtful review. Your constructive suggestions have been truly valuable in helping us improve the clarity and overall quality of the paper.
>
> ***
>
> **[W1&Q1. The latency analysis]**
>
> Thank you for the helpful suggestion. We provide the wall-clock time per image for different generation methods in the table below, measured on an NVIDIA V100 GPU. These methods were trained on the CIFAR-10 dataset. We will include this latency analysis in the revised version of the paper.
> | **Method**  | **DDPM**         | **Score SDE**    | **DDIM**         | **DDIM**           | **EDM**    | **EDM** | **KD**     | **PD**   | **CD (LPIPS)** | **DMD** |
> |-------------|------------------|------------------|------------------|--------------------|------------|---------|------------|---------|----------------|--------|
> | NFE    | 1000             | 2000             | 10               | 100                | 35         | 15      | 1          | 1       | 1              | 1      |
> | Latency | 15.32s            | 45.89s           | 0.51s            | 2.35s              | 0.69s      | 0.30s   | 0.26s      | 0.28s   | 0.02s          | 0.03s  |
> | **Method**  | **1-Rectified flow** | **2-Rectified flow** | **3-Rectified flow** | **2-Rectified flow++** | **CT (LPIPS)** | **iCT** | **iCT-deep** | **ECM** | **HKD** |        |
> | NFE     | 1                | 1                | 1                | 1                  | 1          | 1       | 1          | 1       | 1              |        |
> | Latency | 0.03s            | 0.03s            | 0.03s            | 0.03s              | 0.02s      | 0.02s   | 0.02s      | 0.02s   | 0.03s          |        |
> ***
> **[W2&Q2. Experiments on PIE-Bench]**
>
> Thank you for the valuable suggestion. We would like to clarify that our work does not focus on the text-based image editing task, which PIE-Bench primarily evaluates. As noted in Footnote 1 (line 41) of the manuscript, apart from one-step generation, our study centers on diffusion trajectory control rather than text-guided image editing. Adapting our framework to PIE-Bench would require integrating a text-conditioning branch and retraining the model, which is beyond the current scope. Nevertheless, we agree that this is an interesting and valuable direction and consider it a promising avenue for future work.
>
>
> ***
>
> **[W3&Q3. The proof laid out in Appendix C.2]**
>
> Thank you for your careful reading and for pointing out the unclear parts in Appendix C.2, particularly (14)–(15) and (16)–(18). We apologize for the lack of clarity and will clarify the overall proof in the revised version.
>
> The derivation presented in (14)–(15) of Appendix C.2 relies on two key assumptions to hold. **The assumptions are: (1) the level-based Koopman states $\boldsymbol w_t^{(l)}$ are disentangled across levels labeled by $l$, and (2) the mapping $\mathcal H$ is isotropic w.r.t. the Koopman states**.  Intuitively, this requires that (1) Koopman states at different levels depict different features, and (2) each Koopman state contributes equally to the loss function. A detailed version of the derivation, explicitly accounting for the $o(\cdot)$ terms arising from the assumptions, is
>
> $\mathbb E\_{t\sim\mathcal U[\varepsilon , T), \boldsymbol x\_t\sim p\_\text{data}}\left\Vert \mathcal F\circ\mathcal D\left[\left\\{\textrm e^{-t\boldsymbol A^{(l)}}\mathcal E\_{\boldsymbol \theta}^{(l)}(\boldsymbol x\_t)\right\\}\_{l=1}^L\right]-\mathcal F\circ\mathcal D\left[\\{\boldsymbol w\_0^{(l)}\\}\_{l=1}^L\right]\right\Vert\_2^2$
>
> $\overset\star=\sum\_{l=1}^L\mathbb E\Big\Vert \mathcal H\left[\\{ \textrm e^{-t\boldsymbol A^{(l)}}\mathcal E\_{\boldsymbol \theta}^{(l)}(\boldsymbol x\_t)\text{ at element }l\\}\right]-\mathcal H\left[\\{\boldsymbol w\_0^{(l)}\text{ at element }l\\}\right]\Big\Vert\_2^2+o(\rho )$
>
> $=\sum\_{l=1}^L\mathbb E\ \left\Vert\frac{\partial \mathcal H}{\partial \boldsymbol w^{(l)}}\right\Vert\cdot \Big\Vert \textrm e^{-t\boldsymbol A^{(l)}}\mathcal E\_{\boldsymbol \theta}^{(l)}(\boldsymbol x\_t) - \boldsymbol w\_0^{(l)}\Big\Vert\_2^2+o(\rho +\varepsilon )$
>
> $\overset*=\sum\_{l=1}^L\mathbb E\ \left\Vert\frac{\partial \mathcal H}{\partial \boldsymbol w^{(l)}}\right\Vert\cdot \left\Vert\textrm e^{-t\boldsymbol A^{(l)}}\right\Vert\cdot \Big\Vert \mathcal E\_{\boldsymbol \theta}^{(l)}(\boldsymbol x\_t) - \boldsymbol w\_0^{(l)}\Big\Vert\_2^2 +o(\rho +\varepsilon +\delta )$
>
> where the equations $\overset\star=$ and $\overset*=$ correspond to the two assumptions above, respectively. The error term $o(\rho +\varepsilon +\delta )$ consists of three small-value terms: $\rho$, which measures the extent of entanglement among levels; $\varepsilon$, which quantifies the anisotropy of $\mathcal H$; and $\delta$, which is the range of Koopman spectrum. These error terms reflect the degree to which the assumptions are violated; since our assumptions are generally easy to satisfy in practice, the resulting error is expected to be small.
>
> Regarding Eqs. (16) and (18), we sincerely apologize for the confusion. Eq. (16) already implies the conclusion stated in Proposition C.2, while Eq. (18) provides a corollary that the proposed loss function at $\boldsymbol x_0$ not only provides supervision on the latent-space trajectory, but also infers supervision on $\boldsymbol x_t$, the image-space trajectory. We will remove Eqs. (17)–(18) in the revised version to clarify the proof.

---

### Official Review · Reviewer_MPjE · 2025-06-30

**Clarity:** 3
**Significance:** 3
**Originality:** 4
**Rating:** 5
**Confidence:** 4

**Summary:**

The paper introduces a novel one-step diffusion model based on Koopman operators. Unlike existing few/one-step diffusion models, the proposed Hierachical Koopman Diffusion (HKD) offers several advantages, including improved interpretability during generation and the elimination of iterative sampling for ODE solving.

The HKD framework adopts a hierarchical architecture that projects a noisy input $x_t$ into a set of latents $z_t$ at different spatial resolutions, referred to as Koopman subspaces. The original nonlinear diffusion dynamics of $x_t$ is modeled as a linear evolution within these subspaces, enabling closed-form trajectory solutions and effectively eliminating the need for iterative sampling.

The authors provide a theoretical justification showing that by explicitly modeling diffusion dynamics in a simpler linear form via Koopman operators introduces less error compared to existing one-step diffusion models, which may struggle with learning complex end-to-end one-step mappings.

**Questions:**

1. I wonder whether the hierarchical modeling with varying spatial resolutions from the U-Net architecture is essential to the proposed framework. Can the framework also be applied to DiT-based architectures, given that many recent image foundation models are built on DiT?

2. In the main paper, the encoder is not conditioned on the timestep t, but I saw a description in the appendix that the noise level is fed into the EDM U-Net. I wonder if you've observed any performance changes when not conditioning the encoder on the noise level.

3. Can HKD perform multi-step generation by recursively computing $x_{t-1}$ from $x_t$ through the encoder-decoder pipeline, rather than predicting $x_0$ in a single forward?

**Ethical Concerns:**

["NO or VERY MINOR ethics concerns only"]

**Final Justification:**

I believe this submission presents sufficient novelty and makes a meaningful contribution to the generative modeling literature. Thus, I vote for Accept.

**Limitations:**

The limitations are written in Strengths and Weaknesses.

**Paper Formatting Concerns:**

No formatting issues found.

**Quality:**

3

**Strengths And Weaknesses:**

The paper is well written and easy to follow.

Introducing Koopman theory into the diffusion model literature is novel and theoretically beneficial, opening up a promising new direction for accelerating diffusion models.

Compared to prior work, HKD slightly lags behind iCT but outperforms the other baselines. However, I believe the current results may not reflect optimal hyperparameter settings. The method's novelty outweighs these limitations, and I see strong potential in the proposed approach.

In terms of weaknesses, the paper lacks some key experiments and includes a few claims that appear overstated and would benefit from clarification.

The authors claim that HKD offers better training stability than iCT, but no experimental evidence is provided to show the training stability of HKD. Additionally, the comparison with recent few-step diffusion models is somewhat limited—these baselines are only included on CIFAR-10, and are omitted for the other datasets. Furthermore, the highlighting in the tables does not reflect the best results, which may be misleading to readers.

---

> ### Author Rebuttal · Authors · 2025-07-31
>
> We sincerely thank the reviewer for the encouraging feedback and constructive suggestions, which have been valuable in improving the clarity and quality of our paper. We are especially grateful for the recognition of the novelty, the promising new direction, and the strong potential of our approach.
>
> ***
>
> **[W1. Training stability of HKD]**
>
> Thank you sincerely for pointing this out, and apologize for not including experimental evidence on training stability. We will add such results in the revised version to better support our claim.
>
> From a theoretical perspective, the training stability of HKD is supported by two key aspects of our formulation: (1) The exponential formulation in the Koopman space ensures sufficiently large gradients for spectra with magnitude around 1, which shows superiority in the effectiveness of switching underlying ODEs and helps mitigate the instability issues often encountered in consistency models. Consistency models may suffer from high-variance gradient estimators due to sharp decision boundaries in the ODE flow, making them difficult to train [1]. (2) The supervision of the Koopman trajectory with inputs at multiple time points ensures that the proposed method captures the ideal spectrum for the evolution of modes in a more stable and averaged manner, which enhances the stability.
>
> As experimental evidence, we include the results of training HKD five times independently on the CIFAR-10 dataset. The FID scores and their standard deviation are reported in the table below. We summarize the FID results over every 10 epochs during training across the five runs, demonstrating stable convergence behavior.
>
> |                   | Before Training | Epoch 0    | Epoch 10  | Epoch 20  | Epoch 30  | Epoch 40  | Epoch 50  | Epoch 52  | Epoch 53  |
> | ----------------- | --------------- | ---------- | --------- | --------- | --------- | --------- | --------- | --------- | --------- |
> | FID (Mean ± Std ) | 447.13          | 11.82±0.29 | 5.70±0.21 | 4.71±0.17 | 4.09±0.18 | 3.52±0.11 | 3.31±0.03 | 3.31±0.02 | 3.30±0.02 |
>
> [1] Silvestri, Gianluigi, et al. "Training consistency models with variational noise coupling." *arXiv preprint arXiv:2502.18197* (2025).
>
> ***
>
> **[W2. The comparison with few-step methods on FFHQ]**
>
> We apologize that some recent few-step diffusion models were not included in our evaluation on the FFHQ dataset. This is because these methods were not originally trained or evaluated on FFHQ, leading to the unavailability of official pretrained models and standardized evaluation protocols for this dataset.
>
> Among the main comparison baselines we selected, one is the few-step EDM model, which is a state-of-the-art diffusion backbone. While EDM performs very well with a large number of steps, its performance drops significantly in the few-step regime.
>
> Another key baseline is ECM, a representative one-step consistency model that, like our method, initializes the network weights from a pretrained score model. We chose to compare against ECM under the same configuration to fairly assess our method’s effectiveness on more complex datasets such as FFHQ.
>
> ***
>
> **[W3. The highlighting in the tables]**
>
> We sincerely apologize for the confusion caused by the highlighting in the tables. Our original intention was to draw attention to the results of our proposed method, not to imply that they represent the best performance overall. We will remove the highlighting in the revised version to ensure clarity. Thank you for pointing this out.
>
> ***
>
> **[Q1 (I). The hierarchical modeling with varying spatial resolutions]**
>
> We appreciate the reviewer’s insightful question. While the hierarchical architecture enhances the representational capacity, it is not essential for the theoretical validity or functioning of the proposed Koopman-based framework.  The hierarchical design, inspired by the U-Net architecture, encourages a structured decomposition of modes across different spatial resolutions. This facilitates a frequency-aware partitioning of the feature space and enriches the model's representational capacity by allowing different resolution bands to capture different aspects of the dynamics. Thus, as shown in the ablation study, incorporating hierarchical modeling can improve generation quality.
>
> However, we would like to emphasize that the core mechanism of our framework, i.e., learning a latent representation amenable to Koopman-based evolution, is not strictly dependent on the hierarchical structure. In principle, even without hierarchical modeling, the network has the capacity to learn the necessary information at the final resolution layer. That is, the model will naturally be incentivized to encode the full dynamical modes into the final latent features to enable accurate one-step prediction through the Koopman operator.
>
> **[Q1 (II). Applied to DiT-based architectures]**
>
> Yes, our framework can be applied to DiT-based architectures. Specifically, the input tokens of each DiT block can be used for Koopman evolution. Our Koopman framework only requires a feature representation with sufficient expressiveness to model temporal dynamics and enable one-step generation. As long as the feature extractor, whether a U-Net or a DiT, provides such features, our method remains applicable.
>
> ***
>
> **[Q2. Condition on the time *t*]**
>
> Thank you sincerely for your careful reading and thoughtful observation. We have observed a minor drop in performance when removing noise-level conditioning, particularly in convergence speed. For example, without conditioning, the FID at epoch 50 reaches 3.83, which is slightly higher than the 3.30 achieved when the noise level is explicitly provided. This degradation is expected, as images at different noise levels lie in different feature spaces. Removing the noise level input reduces the encoder’s ability, especially when pretrained, to map noisy inputs into a unified latent space, thereby slightly compromising training efficiency and performance.
>
> ***
>
> **[Q3. Multi-step generation capability]**
>
> Our framework can, in principle, support recursively computing $x_{t-1}$ from $x_t$ through the encoder–Koopman–decoder pipeline. However, this would require modifications to the current training setup. In our work, we focus on one-step generation, where the Koopman model is always trained to evolve features toward a fixed target time $t=0$, and decoding is only performed at this final time step to recover $x_0$. To enable multi-step generation by recursively computing $x_{t-1}$ from $x_{t}$, we would need to adjust both the Koopman evolution targets to perform one-step evolution from $z_{t}$ to $z_{t-1}$ and the decoder to support feature decoding at arbitrary intermediate time steps, not just at $t=0$.

---

> > ### Comment · Reviewer_MPjE · 2025-08-05
> > **Final rating**
> >
> > As the authors have addressed all of my concerns, I recommend 'Accept' as my final rating.

---

> > > ### Author Response · Authors · 2025-08-06
> > >
> > > We sincerely thank the reviewer for the positive feedback and for recommending acceptance. We truly appreciate your time and effort in reviewing our work.

---

### Official Review · Reviewer_q79C · 2025-07-03

**Clarity:** 3
**Significance:** 3
**Originality:** 3
**Rating:** 4
**Confidence:** 4

**Summary:**

This paper proposes Hierarchical Koopman Diffusion (HKD), a framework that leverages Koopman operator theory to enable one-step sampling for diffusion models while preserving interpretable generative trajectories. It also achieves competitive results for one-step generation.

**Questions:**

1.	Do you have results on ImageNet at 64×64 or 256×256 resolutions? It would be helpful to see whether the method scales effectively to larger and more diverse datasets.
2.	How do the extracted features impact the effectiveness of the Koopman operator in this context? It would be valuable to include more discussion on what constitutes good feature extraction for achieving better one-step results.
3.	Is it necessary to extract multi-scale features from images to apply the Koopman operator theory effectively? For better clarity, I suggest including a simple toy example to illustrate how the Koopman framework works in practice — for example, mapping a standard Gaussian to another simple distribution.
4. Does the Koopman framework require pretrained models, or can it be trained entirely from scratch?

**Ethical Concerns:**

["NO or VERY MINOR ethics concerns only"]

**Final Justification:**

The response addressed most of my questions and concerns. I prefer to maintain my score.

**Limitations:**

See weakness and questions.

**Quality:**

3

**Strengths And Weaknesses:**

Strengths:
The paper is well-structured, with a solid theoretical foundation and clear technical novelty. The application of Koopman operator theory to diffusion distillation is particularly interesting. The authors also provide extensive experimental validation, detailed theoretical derivations, and ablation studies.


Weaknesses:
The paper does not include results on larger or more challenging datasets such as ImageNet, leaving the method’s scalability and generalization to more complex data unproven. While the reported results are competitive, they do not reach state-of-the-art performance, raising questions about the practical benefit of introducing the Koopman operator framework compared to existing simpler or more established approaches.

---

> ### Author Rebuttal · Authors · 2025-07-31
>
> Thank you sincerely for your thoughtful feedback. We deeply appreciate the recognition of our work’s theoretical foundation, methodological novelty, and comprehensive experimental validation.
>
> ***
>
> **[W1. Results on larger or more challenging datasets]**
>
> To demonstrate the generalization ability of our framework to larger and more complex datasets, we conducted experiments on the more complex FFHQ dataset at a resolution of 64×64. The results, presented in Table 2, highlight the potential of our framework on higher-resolution and more challenging data.
>
> While we have not yet included results on even larger datasets such as ImageNet, we believe our approach is inherently scalable due to its modular Koopman-based formulation, which allows effective training across tasks of different scales and compatibility with various feature extractors, such as DiT and other network architectures, which enables it to adapt to more complex image generation tasks. Extending our method to larger-scale datasets remains an important and promising direction for future work.
>
> ***
>
> **[W2. The practical benefit of introducing the Koopman operator framework]**
>
> We appreciate the reviewer’s thoughtful comment on the practical utility and comparative advantage of introducing the Koopman operator framework. We would like to clarify that, beyond achieving competitive one-step generation performance, our framework offers additional benefits in terms of interpretability and controllability by preserving the diffusion trajectory, which is generally unavailable in existing one-step generation approaches. This property allows the use of training-free sampling-path control techniques, as shown in Figure 5.
>
> Moreover, our method achieves competitive performance without extensive hyperparameter tuning, highlighting its practical applicability. Compared to existing one-step methods, our framework only introduces an additional Koopman module, which requires learning the matrix $\boldsymbol{A}$. However, the design of this matrix is tridiagonal, meaning it does not significantly increase the number of parameters. From an implementation perspective, once the Koopman evolution function is written, it can be easily integrated into the existing encoder-decoder pipeline with just one additional line of code. Thus, our framework remains both lightweight and effective, building upon established methodologies while introducing minimal additional complexity.
>
> ***
>
> **[Q1. Results on ImageNet at 64×64 or 256×256 resolutions]**
>
> We apologize for not having results on ImageNet at 64×64 or 256×256 resolutions. Due to the computational cost, training on ImageNet 64x64 with 8×A100 GPUs would require more than one week.
>
> In addition to experiments on the CIFAR-10 dataset, we demonstrate the generalization ability of our method on the more complex and higher-resolution FFHQ dataset (64×64) in the manuscript. These results highlight the potential of our framework on more challenging data.
>
> ***
>
> **[Q2 (I). Impact of feature quality on Koopman modeling]**
>
> Thank you for your insightful question. In our framework, the features extracted by the encoder help identify representative and expressive modes (as shown in Figure 4 of the manuscript, where different semantic features are associated with different modes). The Koopman operator focuses on assigning spectral components to these features (or modes) to learn how they evolve over time.
>
> Therefore, the extracted features impact the Koopman framework by influencing the tractability of the identification of the spectrum. **It can be concluded that features with more simplex trends of evolution ease the training of the Koopman operator.** For instance, a good estimation of the diffusion noise would be excellent as its trend is as simple as dropping throughout the evolution. The requirement is mainly caused by the simplification we applied in Appendix C.1 where we formulated the Koopman operator into a tridiagonal matrix. The formulation complicates the depiction of features with multiple dynamics, such as border features containing both semantic evidence and sharp details, which correspond to the middle and high frequency features in Fig. 4(c) and (d), respectively. Having these kinds of features would result in the best spectra to model their evolution, but also a loss in performance.
>
> However, more redundant dimensions in the Koopman space can solve the problem by selecting better features for reconstruction. Therefore, given a sufficient number of Koopman dimensions, the overall result mainly depends on the representativeness of the extracted features, just as any one-step model would do, which leads to the second requirement of good features.
>
> **[Q2 (II). What constitutes good feature extraction for achieving better one-step results]**
>
> Regarding the impact on the overall one-step results, good features require two properties, i.e., **(1) each feature has a simplex evolution trend as in one of the modes visualized in Fig. 4, and (2) features combined are sufficiently representative.** The necessity of the second property can be seen in the rows (a)-(d) in Fig. 4 as missing representative features degrade the image quality. These properties require the encoder to disentangle multiple dynamics and preserve as many faithful image characteristics as possible. The first requirement, as discussed above, improves the effectiveness of the Koopman operator but is harder to achieve as the pre-trained extractor does not initially satisfy the condition. Fortunately, the impact can be tackled by a sufficient number of Koopman dimensions. The second requirement, however, dominates the quality of one-step image generation and requires a better pre-trained model and fine-tuning of hyperparameters.
>
> ***
>
> **[Q3. Necessity of multi-scale features and a simple toy example]**
>
> Extracting multi-scale features is not of great necessity to applying the Koopman theory. As shown in the ablation study, hierarchical modeling improves generation quality by helping organize dynamical modes across levels. However, even without hierarchical modeling, the model can still learn a sufficiently expressive latent representation at a single resolution. This representation captures the full set of dynamical modes required for accurate one-step prediction under Koopman evolution.
>
> Regarding the visualization of the toy example, we will include it in the Appendix to demonstrate the distribution evolution of one pixel during the revision to showcase the evolution estimated by the Koopman framework.
>
> The Koopman-space trajectory of each pixel, evolving from a standard Gaussian to its true data distribution, closely mirrors the local frequency spectral evolution patterns shown in Figure 4(e).
>
> In this figure, we use a tinted background to illustrate the distribution evolution from time $T$ to $0$ where the color band at time $t$ represents the momentary probability density of a Koopman state. The density function is estimated by the histogram of a certain element in the state from 1000 randomly sampled images. In the foreground, we plot the trajectories corresponding to the values of this element using the Koopman framework.
>
> ***
>
> **[Q4. Pretraining dependency and trainability from scratch]**
>
> The Koopman framework can be trained entirely from scratch. Initializing with a pretrained model is solely intended to improve training efficiency and accelerate the convergence of our model. Specifically, on the CIFAR-10 dataset, training with a pretrained model takes only 2–3 days on 8×V100 GPUs, whereas training from scratch requires approximately one week. This efficiency gain arises because the pretrained model provides the Koopman module with reasonably good features at the early stage of training, which speeds up the learning process. However, if given sufficient training time, the encoder can still learn to extract representative features without any pretraining.

---

### Official Review · Reviewer_Vr9m · 2025-07-03

**Clarity:** 3
**Significance:** 2
**Originality:** 3
**Rating:** 4
**Confidence:** 4

**Summary:**

This paper introduces Hierarchical Koopman Diffusion (HKD) to bridge the gap between fast, one-step generative models and the interpretable, controllable trajectories of traditional diffusion models. It is based on the Koopman operator theory to linearize the nonlinear diffusion ODE dynamics.
The authors propose a U-Net-like architecture to encode a noisy image x_t into a hierarchical set of latent observables, which are then evolved in a single step via a learned linear Koopman operator. A corresponding decoder then maps these evolved latents back to the image space. This paper shows that this framework not only enables one-step generation but also provides interpretability through spectral analysis of the learned Koopman operators. The paper conducts experiments on CIFAR-10 and FFHQ, demonstrating competitive FID scores.

**Questions:**

- How critical is the initialization from a pre-trained EDM model (U-Net)?
- The authors argue against supervising in the latent space because it "reduces training flexibility and introduces gradients that poorly reflect perceptual discrepancies" (line 158). Can you please provide more evidence?
- Could one not achieve similar frequency-based editing by performing a Fourier transform on the feature maps at different levels of a standard U-Net-based one-step model and manipulating those coefficients?

**Ethical Concerns:**

["NO or VERY MINOR ethics concerns only"]

**Final Justification:**

The response addressed my concerns. I'd maintain my score and recommend acceptance.

**Limitations:**

yes

**Quality:**

2

**Strengths And Weaknesses:**

- The application of Koopman operator theory to accelerate diffusion model sampling is a novel direction.
- One of the strengths is the successful demonstration of interpretability. The spectral analysis in Figure 4, showing a correspondence between eigenvalues and semantic image structures, is interesting.
- HKD achieves competitive FID scores for a one-step generator on both CIFAR-10 (3.30) and FFHQ (5.70).

---

> ### Author Rebuttal · Authors · 2025-07-31
>
> We sincerely thank the reviewer for the positive feedback and for recognizing the novelty, interpretability, and performance of our approach.
>
> ***
>
> **[Q1. Importance of the initialization from a pre-trained EDM model]**
>
> The initialization from a pre-trained EDM model is important for improving training efficiency of our framework, as it provides well-structured feature representations in the early stages, which facilitates faster convergence. However, this initialization is not essential to the framework’s design: our model can be trained from scratch and the pre-trained model is used purely to reduce training time. Specifically, on the CIFAR-10 dataset, training with a pretrained model takes only 2–3 days on 8×V100 GPUs, whereas training from scratch requires approximately one week.
>
> ------
>
> **[Q2. Justification against supervision in the latent space]**
>
> We apologize for the confusion. We will clarify this point more explicitly in the revised version. The LPIPS study [1] provides evidence that accurately capturing perceptual discrepancies requires a carefully designed and fixed feature extractor. Supervising in the Koopman space with fixed feature extractor $f$ (e.g., VGG) is equivalent to applying a perceptual loss under the extractor $f\circ\mathcal E$, where $\mathcal E$ is trainable encoder. This training of $\mathcal E$ lacks the perceptual alignment of pretrained perceptual encoders, leading to gradients that may poorly reflect actual perceptual differences in image space.
>
> Furthermore, enforcing supervision in the latent space restricts the encoder's ability to adapt freely to downstream tasks. It may hinder the model’s capacity to learn more expressive or task-specific features. For these reasons, we opt for supervision at the image level, which allows the encoder to evolve flexibly and ensures better alignment with perceptual quality.
>
> [1] Zhang, Richard, et al. "The unreasonable effectiveness of deep features as a perceptual metric." *Proceedings of the IEEE conference on computer vision and pattern recognition*. 2018.
>
> ***
>
> **[Q3. Achieving similar frequency-based editing via a Fourier transform on the feature maps]**
>
> Yes, this is a very interesting idea. In principle, we could achieve frequency-based editing by applying Fourier transforms to feature maps at different levels of a standard U-Net-based one-step model and manipulating the corresponding frequency coefficients.
>
> However, directly injecting high-frequency components without an intermediate evolution process may lead to distortions, because even small spatial misalignments between the source and target images can disrupt the high-frequency details. In contrast, the Koopman-based spectral decomposition provides a structured intermediate evolution, which helps preserve fine details and ensures more stable frequency-level control.

---

### Note · Authors · 2025-08-12

Our work introduces a novel framework leveraging Koopman operator theory to accelerate diffusion model sampling. This approach enables **one-step generation** while **preserving the diffusion trajectory**, offering **interpretability** and **controllability** not present in existing one-step methods.

We thank all reviewers for recognizing the **novelty and potential of our approach**: MPjE and Vr9m noted that our method opens up a promising new direction for accelerating diffusion models, while xGGM emphasized its potential for broader impact. Vr9m further commended our successful demonstration of interpretability. Experimentally, Vr9m observed that our method achieves competitive results for one-step generation, and q79C acknowledged that we provide **extensive experimental validation**. Theoretically, q79C further recognized our **solid theoretical foundation** and detailed derivations.

We addressed the reviewers' concerns with targeted clarifications and additional results. Following xGGM’s suggestion, we added a **latency analysis** comparing wall-clock times across methods for both generation and editing scenarios, and clarified the **assumptions used in the proof in Appendix C.2**, particularly between lines 14–15, when establishing equivalence. For MPjE’s concern that our **training stability** may have been overstated, we provided both theoretical justification and additional experiments, fully resolving all of their concerns. Regarding **scalability**, in response to q79C's comments on the lack of results on larger or more challenging datasets such as ImageNet, we clarified that experiments on the complex **FFHQ 64×64** dataset (Table 2) already demonstrated the framework’s **generalization potential** to higher-resolution and more challenging data. On the q79C's point that our results are competitive but not SOTA, we clarified that beyond competitive one-step generation performance, our trajectory-preserving property offers **unique interpretability and controllability advantages**, enabling training-free sampling-path control (Figure 5), which is generally unavailable in existing one-step methods.

In summary, we have addressed all major concerns while underscoring our method’s **novelty, strong theoretical grounding, competitive empirical performance, and unique interpretability benefits**.

---

### Decision · Program_Chairs · 2025-09-17

**Decision:**

Accept (poster)

**Comment:**

This work proposes a fast inference algorithm for diffusion model based on the Koopman theory. The Koopman theory is a framework that lifts a nonlinear dynamics to a linear dynamics. In this paper, Koopman theory is leveraged to approximate the nonlinear backward diffusion with a linear system in a feature space. Once this feature space is learned, fast inference can be achieved by solving this linear system in closed form. In addition to efficiency, the authors claim interoperability is another benefit of the proposed method. All reviewers agree that applying Koopman theory to diffusion model sampling is a novel and interesting idea. Several weaknesses are pointed out including the scalability of the algorithm to high resolution images. Most of comments have been properly addressed in the rebuttal as indicated in final justification. Overall, this is an interesting contribution to diffusion generative models.